# Exploring Immune Redox Modulation in Bacterial Infections: Insights into Thioredoxin-Mediated Interactions and Implications for Understanding Host–Pathogen Dynamics

**DOI:** 10.3390/antiox13050545

**Published:** 2024-04-29

**Authors:** Omer M. A. Dagah, Billton Bryson Silaa, Minghui Zhu, Qiu Pan, Linlin Qi, Xinyu Liu, Yuqi Liu, Wenjing Peng, Zakir Ullah, Appolonia F. Yudas, Amir Muhammad, Xianquan Zhang, Jun Lu

**Affiliations:** 1Engineering Research Center of Coptis Development and Utilization/Key Laboratory of Luminescence Analysis and Molecular Sensing, Ministry of Education, College of Pharmaceutical Sciences, Southwest University, Chongqing 400715, China; omer20062010@gmail.com (O.M.A.D.); brysonbilton@gmail.com (B.B.S.); 1843476a@gmail.com (M.Z.); qp0721tan@outlook.com (Q.P.); qll199901@outlook.com (L.Q.); lxy8980@outlook.com (X.L.); lyq20001121@outlook.com (Y.L.); l17323908707@outlook.com (W.P.); zakirktk1996@gmail.com (Z.U.); appoloniafulgence@gmail.com (A.F.Y.); amirmkgarba@gmail.com (A.M.); 2Hygeia Hospital, Chongqing 401331, China; xqzhng@hotmail.com

**Keywords:** bacterial infections, pattern recognition receptors (PRRs), immune redox modulation, oxidative stress, thioredoxin (Trx), TXNIP, immune modulatory peptides

## Abstract

Bacterial infections trigger a multifaceted interplay between inflammatory mediators and redox regulation. Recently, accumulating evidence has shown that redox signaling plays a significant role in immune initiation and subsequent immune cell functions. This review addresses the crucial role of the thioredoxin (Trx) system in the initiation of immune reactions and regulation of inflammatory responses during bacterial infections. Downstream signaling pathways in various immune cells involve thiol-dependent redox regulation, highlighting the pivotal roles of thiol redox systems in defense mechanisms. Conversely, the survival and virulence of pathogenic bacteria are enhanced by their ability to counteract oxidative stress and immune attacks. This is achieved through the reduction of oxidized proteins and the modulation of redox-sensitive signaling pathways, which are functions of the Trx system, thereby fortifying bacterial resistance. Moreover, some selenium/sulfur-containing compounds could potentially be developed into targeted therapeutic interventions for pathogenic bacteria. Taken together, the Trx system is a key player in redox regulation during bacterial infection, and contributes to host–pathogen interactions, offering valuable insights for future research and therapeutic development.

## 1. Introduction

Pathogenic bacteria prompt a sophisticated defense mechanism in the host, characterized by a finely tuned balance of inflammatory mediators and redox processes. In response to bacterial invasion, the immune system releases inflammatory cytokines, which then activate redox-sensitive signaling pathways. These pathways play a vital role in orchestrating an effective immune defense, highlighting the dynamic and intricate connection between bacterial populations and host health [1,2]. The immune response to bacterial infections relies on various immune cells, each with its own specialized functions and capabilities. For example, neutrophils, among the initial immune responders, swiftly migrate to infection sites to engulf and eliminate pathogens [3,4,5]. Macrophages, versatile immune cells, can be polarized into distinct M1 and M2 subsets, each serving specific functions in either promoting inflammation or resolving it, depending on the microenvironment [6,7,8,9,10]. T cells are instrumental in orchestrating cell-mediated immunity by producing pro-inflammatory cytokines. Dendritic cells bridge the innate and adaptive immune responses, while myeloid-derived suppressor cells (MDSCs) exert immunosuppressive functions to balance inflammation [11,12,13]. Natural killer (NK) cells are capable of both direct cytotoxicity and the release of cytokines to combat infections [14].

The immune response is orchestrated by a complex interplay of pathways and molecules, with nuclear factor kappa-light-chain-enhancer of activated B cells (NF-κB) and mitogen-activated protein kinase (MAPK) serving as central regulators [15]. NF-κB, which is vital for controlling genes linked to inflammation and immune response, is activated by diverse stimuli, including oxidative stress that triggers the production of ROS. These ROS, extending beyond mere metabolic byproducts, also function as signaling entities in immune regulation, activating both the NF-κB and MAPK pathways, and thus activating the NLR family pyrin domain containing 3 (NLRP-3) inflammasome. Its activation catalyzes the recruitment of apoptosis-associated speck-like protein containing a CARD (ASC) and the activation of caspase-1, which is crucial for converting pro-inflammatory IL-1β into its active form. This cytokine, alongside others such as TNF-α and IFNs released following NF-κB activation, plays a pivotal role in inflammation, highlighting the intricacy and connectivity of these immune regulatory pathways [16,17,18,19]. In essence, the coordinated actions of NF-κB, MAPK, NLRP-3, and ROS epitomize the sophisticated regulatory mechanisms that govern the immune system’s response to diverse challenges.

The thioredoxin antioxidant system, comprising thioredoxin (Trx), thioredoxin reductase (TrxR), and nicotinamide adenine dinucleotide phosphate (NADPH), is a cornerstone in maintaining intracellular redox balance and protecting cells from oxidative stress. Utilizing NADPH, TrxR sustains Trx in a reduced state. Trx, encoded by TXN, is positively regulated, while thioredoxin interacting protein (TXNIP) acts as a negative regulator for this antioxidant system. Mammals have three TrxR isoforms (cytoplasmic TrxR1, mitochondrial TrxR2, and testis-specific TrxR3) and two Trx isoforms (cytoplasmic Trx1 and mitochondrial Trx2) [20]. Simultaneously, the GR/GSH/GRX system, reliant on NADPH, contributes to redox homeostasis. Recent studies highlight their importance in immune cells, suggesting potential therapeutic avenues for immune dysfunction and diseases [21].

The aim of this review is to delve into the intricate role of the Trx system in modulating host–pathogen interactions, particularly in the context of bacterial infections. We examine how immune cells leverage redox signaling pathways, including those mediated by ROS/RNS, to fine-tune their functional responses and maintain immune balance. This involves a detailed exploration of the mechanisms through which the Trx system facilitates the initiation and regulation of immune responses, emphasizing its role in both promoting defense mechanisms via thiol-dependent redox regulation across various immune cells (such as neutrophils, macrophages, T cells, dendritic cells, MDSCs, and NK cells) and serving as a strategic target for pathogenic bacteria aiming to evade the immune defense. Furthermore, we will explore how the thioredoxin and glutaredoxin systems significantly contribute to bacterial resilience against oxidative stress and immune responses. By reducing oxidized proteins and adjusting redox-sensitive signaling pathways, these systems boost the survival and aggressiveness of pathogenic bacteria. We also discuss the significance of selenium/sulfur-containing compounds in developing targeted therapeutic interventions against pathogenic bacteria. By highlighting the dynamic interplay between redox processes and immune cells in the context of bacterial infections, this review aims to illuminate the potential for novel therapeutic strategies that target these mechanisms. Through such detailed examination, we aspire to contribute valuable insights into the significance of the Trx system and redox regulation in defending against bacterial infections and to pave the way for future research and therapeutic innovations in immune response optimization.

## 2. Inflammatory Response in Bacterial Infection

The immune response to bacterial infections begins with the identification of pathogen-associated molecular patterns (PAMPs) and damage-associated molecular patterns (DAMPs). PAMPs, which include microbial molecules such as lipopolysaccharides, lipoproteins, and nucleic acids, along with DAMPs, released from injured or dying cells, include molecules like mitochondrial DNA (mtDNA), heat shock proteins (HSP), interleukin 1 alpha (IL-1α), high mobility group box 1 (HMGB-1), and interleukin 33 (IL-33). These initiators are crucial for alerting the immune system to both infection and cellular damage [22]. Detection of these patterns is mediated by pattern recognition receptors (PRRs) located on/in host cells. Among these receptors, toll-like receptors (TLRs) and nucleotide-binding oligomerization domain-like receptors (NLRs) are particularly important (Figure 1) [22]. Toll-like receptors (TLRs), including TLR-1, TLR-2, TLR-4, TLR-5, and TLR-6, are characterized by leucine-rich repeat motifs and are critical components of the innate immune system. These TLRs are expressed on various cell types, including but not limited to macrophages [23,24]. NLRs, including subtypes Nod1 and Nod2, are predominantly found in the cytoplasm of immune and epithelial cells, and are integral to phagocytes like macrophages and neutrophils [25].

Upon activation, TLRs lead to the recruitment of IL-1 receptor-associated kinases (IRAKs) and TNF receptor-associated factor 6 (TRAF-6). This recruitment subsequently activates the TGF-β-activated kinase 1/TAK-1-associated binding protein (TAK-1/TAB-1) complex. The activation of this complex triggers the IKK (IκB kinase) complex, resulting in the phosphorylation and degradation of IκB (inhibitor kappa B) and the release and nuclear translocation of NF-κB to upregulate inflammatory genes [16,24,26]. TLR signaling also stimulates NOX to produce ROS and TRAF-6 enhances mitochondrial ROS (mROS) production, both of which amplify NF-κB signaling and activate the MAPK pathway [19,27,28]. On the other hand, NOD-1 and NOD-2 detect intracellular bacterial components, leading to the recruitment of receptor-interacting serine/threonine-protein kinase 2 (RIPK-2), which also facilitates the activation of the TAK-1/TAB-1 and IKK complexes, similarly resulting in NF-κB activation and nuclear translocation to stimulate immune gene transcription [16,24,26]. NOD pathways further induce ROS production through NADPH oxidase and NOD-2 specifically enhances mROS production, contributing to NF-κB/IKK and MAPK pathway activation [19,27,28].

Central to this mechanism is the activation of the transcription factor NF-κB, which occurs during oxidative stress following the dissociation of Trx from TXNIP. The activation of NF-κB triggers the assembly of NLRP-3 and promotes the expression of numerous inflammatory cytokines, such as TNF-α, various ILs, and IFNs. These cytokines play a crucial role in the inflammatory cascade and are essential for regulating the immune response, which is critical for the effective containment and eradication of bacterial pathogens [17,18,29]. Thus, Trx not only reacts to oxidative stress but also acts as a bridge connecting this stress to the activation of inflammatory pathways, highlighting its essential function in cellular defense mechanisms. These inflammatory mediators trigger vasodilation, which allows plasma proteins and neutrophils to escape from blood vessels, thereby facilitating the migration and activation of macrophages at the infection site [30,31]. Pathogenic bacteria are then engulfed within phagosomes by neutrophils and macrophages. This interaction initiates an oxidative burst, driven by NOX-2, which produces crucial reactive oxygen, nitrogen, and chlorine species. These reactive species are essential in effectively neutralizing invading pathogens [32]. Concurrently, ROS and RNS are produced by immune cells including macrophages, neutrophils, NK cells, and dendritic cells in response to inflammatory signals and microbial components. These reactive molecules are essential for controlling bacterial proliferation and managing inflammatory responses, thereby establishing a loop that perpetuates a continuous cycle of inflammation during infections [33].

## 3. Immune Cells in Bacterial Infections

### 3.1. Neutrophils

Neutrophils, phagocytic white blood cells, play a role in immune responses by being rapidly recruited to sites of tissue injury and infection. This recruitment is facilitated through specific receptors that detect microbial patterns or endogenous DAMPs [34,35]. Neutrophils combat pathogens through processes like phagocytosis, respiratory burst, and the formation of neutrophil extracellular traps (NETs) [4,5].

During phagocytosis, pathogens are engulfed into phagosomes, where neutrophils unleash a respiratory burst by mobilizing azurophilic granules (Figure 2). These are specialized lysosomes packed with additional antimicrobials, including myeloperoxidase (MPO), elastase, proteases, nucleases, peptides, lysozyme, and cathepsin. These granules fuse with the phagosome, thus creating an environment ideal for combating the pathogen [3,4,5,36,37,38]. At the same time, the respiratory burst, or oxidative burst, begins. This critical process involves the activation of the NOX complex on the phagosomal membrane, leading to the generation of O_2_^−^. These superoxide anions are quickly converted to H_2_O_2_ by superoxide dismutase (SOD) also present within the phagosome. The hydrogen peroxide serves as a substrate for MPO, facilitating the formation of hypochlorous acid (HOCl), a potent antimicrobial agent [5,39]. This cascade of reactions significantly enhances the neutrophils’ microbial killing capacity, working synergistically with the enzymatic action of elastase, cathepsin, and other antimicrobial agents to degrade bacterial proteins, nucleic acids, and cell walls, effectively destroying the pathogen. This coordinated sequence of events demonstrates the dynamic and powerful nature of neutrophils in the immune response.

NETs are structures formed by deconstructed DNA and antimicrobial proteins that help neutrophils capture and eliminate pathogens while protecting host tissues [40]. Suicidal NETosis is a process of neutrophil death that involves the activation of NADPH oxidase, neutrophil elastase (NE), and MPO, leading to the release of NETs in response to various stimuli such as phorbol myristate acetate (PMA), IL-8, and lipopolysaccharides (LPS) (Figure 3) [41]. The activation of PKC-α can be initiated by various agents such as PMA, which directly stimulates PKC-α. Other initiators such as the calcium ionophore A23187 and Ca^2^⁺-raising antibodies increase intracellular calcium levels, subsequently activating PKC-α [42]. Additionally, the chemokine IL-8 serves as an initiator by binding to and activating its receptor, C-X-C chemokine receptor (CXCR), which promotes further downstream signaling [43]. Key to the activation of downstream signaling molecules are receptors such as the fibroblast growth factor receptor (FGFR) and the CXCR. FGFR triggers the activation of phospholipase C gamma (PLC-γ), which produces diacylglycerol (DAG) to enhance PKC-α activity. CXCR, activated by IL-8, initiates the phosphoinositide 3-kinases/protein kinase B (PI3K/PKB) signaling pathway and thus activates the transcription factor NF-κB [43,44].

PKC-α triggers a cascade of biochemical reactions that lead to the activation of the RAF/MEK/ERK signaling pathway. This pathway’s activation stimulates NOX, which increases the production of ROS, commonly produced during cellular stress. These elevated ROS levels promote the oxidative dissociation of Trx from its inhibitor TXNIP, subsequently activating the transcription factor NF-κB—a key player in inflammatory responses that can also be activated directly through TXNIP [18,42,45,46]. Additionally, ROS are crucial for the release of azurophilic granules, which are essential for the nuclear processes involved in suicidal NETosis [47]. Once dissociated, Trx, independent of its oxidoreductive capacity, activates PAD-4 [42,45,46]. PAD-4 is an enzyme that converts arginine residues to citrulline in various protein substrates and plays a central role in the formation of NETs by mediating histone citrullination, thereby promoting NET formation in both forms of NETosis [48,49]. Furthermore, increased intracellular calcium levels, triggered by the calcium ionophore A23187 and certain antibodies, directly activate PAD-4 [42]. Additionally, the activation of NF-κB initiates the production of iNOS, which is vital for generating NO [50,51]. This not only boosts inflammatory responses but also establishes a feedback loop that aggravates suicidal NETosis and amplifies the overall inflammatory environment.

In contrast to suicidal NETosis, vital NETosis occurs independently of NADPH oxidase and allows neutrophils to release NETs without undergoing cell death. It can be induced by certain bacteria, including *E. coli* and *S. aureus,* as well as bacterial-specific molecular patterns that are recognized by PRRs [41,52]. Ca^2+^ elevation through pathways like TLR-2, TLR-4, and the complement receptor triggers PAD-4-mediated histone citrullination, histone decondensation, and NET release from vesicles without membrane rupture [42].

Collectively, NETosis operates through distinct suicidal and vital pathways, with the suicidal pathway being notably driven by the oxidative stress-mediated dissociation of Trx from TXNIP. This interaction triggers PAD-4 activation and subsequent histone citrullination, crucial for NET formation. These insights provide a foundation for developing therapeutic interventions aimed at controlling inflammation and improving immune responses in disease states.

### 3.2. Macrophages

Macrophages maintain tissue balance and respond to inflammation, serving as essential immune cells present in tissues or migrating to sites of infection or injury [6,7]. Macrophage polarization involves the distinct functional responses of macrophages to specific microenvironmental signals, resulting in their differentiation into two primary subsets: M1 and M2 macrophages [8]. M1 macrophages are typically induced by stimuli such as lipopolysaccharide (LPS) and Th1 cytokines, notably interferon gamma (IFN-γ). These initiators bind to their respective receptors, including TLR (for LPS) and IFN-γ receptors, which activate signaling intermediates such as the Janus kinase-signal transducer and activator of transcription 1 (JAK-STAT-1) and NF-κB pathways [9,53].

Redox regulation is crucial for the functionality of M1 macrophages. ROS produced by NOX-2 are key elements that maintain inflammation, enable phagocytosis, and facilitate cytotoxic actions for pathogen elimination, while also supporting crucial autophagy processes. Notably, while NOX-2-derived ROS are essential for inducing autophagy, mutations that impair superoxide formation can lead to hyperinflammation and recurrent infections [54,55,56]. The TXNIP/NF-κB/NLRP-3 inflammasome-caspase-1 pathway plays a crucial role in regulating phagosomal pH and bacterial clearance in macrophages (Figure 4). Upon bacterial engulfment, NOX-2 generates ROS. This triggers the dissociation of Trx from TXNIP, subsequently activating NF-κB along with NLRP-3. Caspase-1 is then activated, serving a dual purpose by processing cytokines and regulating ROS levels. This dual function contributes to the maintenance of an acidic phagosomal environment (pH 4) during the late stage of phagosome maturation by inhibiting NADPH oxidase activity. This controlled acidity enhances the activity of acidic enzymes, promoting bacterial degradation and effective immune responses [17,24,29,57].

NO, produced by inducible iNOS, demonstrates varied effects based on its concentration within M1 macrophages. At low levels (1–30 nM), NO exhibits anti-inflammatory properties. Intermediate levels (30–100 nM) stabilize hypoxia-inducible factor 1 alpha (HIF-1α), crucial for various aspects of macrophage functionality including metabolism and viability. Higher concentrations (approximately 400 nM) promote cytotoxicity, while excessive levels (>400 nM) trigger apoptosis, assisting in the resolution of inflammation and promoting the proliferation of M2 macrophages, essential for wound healing [58,59,60].

In contrast to M1 macrophages, M2 macrophages polarize in response to various Th2 cytokines, most notably IL-4 and IL-13. This activation leads M2 macrophages to display an anti-inflammatory profile, generating anti-inflammatory cytokines like IL-10 and transforming growth factor beta (TGF-β) [9]. The transition from M1 to M2 macrophages marks a critical shift in cellular redox homeostasis and involves several key biochemical pathways critical for tissue remodeling and wound healing. The process begins with the activation of peroxisome proliferator-activated receptor gamma 1 (PPAR-γ) by IL-4, mediated via STAT-6 signaling. This activation results in the suppression of PKC-α, which in turn inhibits NOX-mediated ROS production via the RAF/MEK/ERK pathway (Figure 5) [61,62].

Simultaneously, IL-4 downregulates NOX-2 activity, directly promoting the expression of cathepsins S and L, which enhances the reductive capacity of the macrophage lumen, thereby increasing the efficiency of protein degradation within phagosomes. These changes are facilitated by a reduction in NOX-2-mediated disulfide bond formation, optimizing the macrophages for improved wound repair and tissue remodeling [70,71]. Additionally, IL-4 boosts arginase-1 (Arg-1) activity, which shifts the cellular metabolism from producing nitric oxide (NO) via inducible nitric oxide synthase (iNOS) to producing ornithine, a precursor for collagen and other essential components for tissue repair, via a STAT-6/PPAR-γ-dependent mechanism [63,72,73,74]. The reduction in iNOS activity and NO levels further enhances the M2 phenotype [64,75].

In parallel, prostaglandin E2 (PGE-2) activates PI3K/PKB pathway, contributing to the suppression of NOX-2 activation and supporting the M2 polarization process [69]. The redox-sensitive transcription factor Nrf-2 also plays a crucial role; it regulates the expression of Cu, Zn-superoxide dismutase (SOD-1), and Trx systems [76,77,78]. SOD1, through the generation of H_2_O_2_, aids in activating STAT-6 and interferon regulatory factor 4 (IRF-4), crucial for initiating M2-specific gene transcription [65,66]. Moreover, the overexpression of SOD-1 results in a decrease in iNOS gene expression and a reduction in NO synthesis, while concurrently promoting arginase I expression and augmenting urea generation. These changes impact collagen synthesis and contribute to the development of fibrosis [65,66]. This transition is dependent on the cell’s redox state, with a finely tuned balance of ROS being essential for effective M2 differentiation [79]. Nrf-2 further induces the expression of Trx/Prx and related enzymes by binding to antioxidant response elements (AREs) in their promoters, augmenting the cells’ defenses against oxidative stress and maintaining redox equilibrium [67,68].

### 3.3. Dendritic Cells (DCs)

Dendritic cells (DCs) are key immune cells that bridge innate and adaptive immunity and are classified into two main types based on their functional and phenotypic characteristics: plasmacytoid DCs (pDCs) and conventional DCs (cDCs). These cells are crucial for both tumor immunotherapy and the management of autoimmune diseases [11,12,80,81]. Both pDCs and cDCs heavily utilize intracellular ROS, including H_2_O_2_, to modulate their antigen-presenting functions and survival. Elevated H_2_O_2_ levels disrupt cellular functions by interfering with tyrosine phosphatase activity, potentially leading to DC apoptosis. This exemplifies the critical balance maintained by DCs in utilizing ROS for cellular processes while avoiding detrimental effects [82,83,84,85,86].

The R837 ligand of TLR7 plays a significant role in pDC regulation. Interaction with this receptor inhibits the activation and maturation of pDCs, subsequently affecting their ability to secrete cytokines. This modulation is essential for controlling immune responses, particularly in pathogenic infections or inflammatory conditions [86]. Key to DC function are the NADPH oxidases, NOX-2 and NOX-5, which regulate the generation of ROS. NOX-2 influences phagosomal pH and antigen presentation and modulates the inflammatory response by inhibiting IL-12 production via the p38-MAPK pathway. Concurrently, mROS, produced by NOX-5 and the mitochondrial subunit P22phox, are involved in the differentiation of cDCs via the JAK/STAT/MAPK and NF-κB signaling pathways. Furthermore, these mROS also enhance cytokine secretion, emphasizing the complex interactions between different ROS sources within dendritic cells [87,88,89].

pDCs are notable for their production of type I interferons (IFN-α and IFN-β), which are integral to tumor defense and immune regulation [82,83,84]. The production of IL-12, a key proinflammatory cytokine, is intricately controlled by ROS through the p38-MAPK pathway, illustrating the main role of ROS in immune modulation [88,90,91]. Additionally, DCs secrete Trx1, which promotes T-cell proliferation and modulates T-cell receptor signaling. This is discussed in much more detail later [92].

### 3.4. T Cells

T cells are crucial elements of the adaptive immune system, mainly involved in cell-mediated immune defenses. They derive from hematopoietic stem cells in the bone marrow and mature in the thymus, which is why they are called T cells [93]. Among the various types of T cells, helper T cells (Th) play a central role in orchestrating the immune response. They achieve this by producing cytokines that guide the behavior of other immune cells. Helper T cells are subdivided into several groups including Th1, Th2, Th17, and Tfh cells, each tailored to support specific immune functions [94].

T-cell differentiation is profoundly influenced by cytokines in the local environment; specifically, IL-4 promotes the formation of the Th2 subset, while IFN-γ encourages Th1 development. This targeted response equips T cells to effectively address a wide variety of immune challenges (Figure 6) [95,96]. T-cell differentiation critically depends on antigen presentation by DCs. These cells initiate immune responses by interacting with naive T cells, a process that is essential for activating the T cells and steering their differentiation into various helper T-cell subsets [92]. Trx1, a protein secreted by dendritic cells, further regulates this differentiation by adjusting the expression of important T cell receptors such as CD4 and CD30. It also facilitates the conversion of cystine to cysteine inside cells, which is vital for T-cell proliferation [92]. However, this necessary conversion is impeded by myeloid-derived suppressor cells (MDSCs), which sequester cystine and limit its availability to antigen-presenting cells, thereby hindering T-cell functionality and enhancing immune suppression [97,98]. Moreover, the maturation of dendritic cells themselves is influenced by NK cells, which release TNF-α and IFN-γ. IFN-γ not only matures DCs but also significantly drives the differentiation of Th1 cells, making more intricate the network of immune cell interactions and responses [99,100].

The T-cell receptor (TCR) is a key signaling intermediate that propagates activation signals within the cell. Upon TCR activation, there is an increase in intracellular production of H_2_O_2_, which is necessary for activating various transcription factors including NF-κB. This factor promotes the transcription of IL-2 and the IL-2 receptor alpha chain gene, which are critical for T-cell activation and proliferation. NF-κB also drives the expression of inflammatory cytokines that enhance the immune response [101,102]. Moreover, a loop is formed between Trx and IFN-γ; Trx-1 boosts the production and secretion of IFN-γ in Th1 cells, and the heightened levels of IFN-γ subsequently elevate Trx1 levels. Additionally, IFN-γ-activated Trx-1 within macrophages enhances the secretion of the Th1 cytokine IL-12 by modulating the thiol redox state. Moreover, extracellular Trx1 specifically deactivates the cytokine function of IL-4, thereby suppressing the Th2 immune response [92].

### 3.5. Myeloid-Derived Suppressor Cells (MDSC)

MDSCs represent a group of immature myeloid cells with potent immunosuppressive capabilities [13]. They encompass two primary categories, namely, polymorphonuclear MDSCs (PMN-MDSCs) and monocytic MDSCs (M-MDSCs) [13,103]. The activation and function of MDSCs are primarily initiated by bacterial infections, which can cause either exaggerated inflammation or sustained production of pro-inflammatory cytokines and chemokines. An important initiator of MDSC function is IL-6, a cytokine that specifically triggers the JAK-2/STAT-3 signaling pathway within MDSCs. This pathway is crucial for their role in immunosuppression through the production of ROS [13,103,104,105,106,107]. The JAK-2/STAT-3 pathway, activated by IL-6 in MDSCs, upregulates NOX-2, a key component of NADPH oxidase responsible for ROS generation. This increase in ROS production is crucial for the immunosuppressive functions of MDSCs. Additionally, STAT-3, a vital transcription factor in this pathway, enhances the transcription of genes that further elevate ROS levels, amplifying MDSCs’ ability to suppress immune responses [104,105,106,107]. MDSCs are equipped with specific surface receptors, including cystine transporters, that are pivotal for their function. These transporters import cystine from the extracellular environment, crucial for modulating its availability to other immune cells such as dendritic cells and macrophages. By controlling cystine uptake and restricting its conversion to cysteine, MDSCs effectively prevent T-cell activation (see Figure 6 above). The absence of enzymes like cystathionase and the alanine–serine–cysteine (ASC) transporter emphasizes their reliance on imported cystine for cysteine production, underlining their strategic role in immune regulation by modulating cysteine availability during antigen presentation [97,98]. Additionally, beyond generating ROS, MDSCs employ mechanisms like lipid oxidation and the expression of antioxidants to regulate ROS levels, highlighting their complex role in immune system modulation [104,105,106,107].

### 3.6. Natural Killer (NK) Cells

Natural killer (NK) cells are a type of granular lymphocyte that play a crucial role in the immune response against pathogens. These cells express specific surface receptors that allow them to detect and destroy infected cells or cancer cells. NK cells operate through both direct and indirect mechanisms to combat microbial threats. They can induce apoptosis in target cells by engaging with surface ligands and also contribute to immune defense by releasing cytokines such as IFN-γ and tumor necrosis factor-alpha (TNF-α). These cytokines activate other immune cells, including macrophages, enhancing the overall immune response [108].

Cytokines like IL-2 and IL-12 are vital for the function and activation of NK cells. IL-2 promotes the lytic activity of NK cells against pathogens, whereas IL-12 enhances their overall activity and stimulates the production of IFN-γ, which is critical for controlling pathogen growth [108]. IL-15 is another crucial cytokine for NK cell function; it activates pathways such as JAK-STAT-5 and PI3K/PKB, which are essential for NK cell development and maturation. Additionally, IL-15 engages the Trx system, which plays a role in reducing intracellular levels of ROS, thus maintaining cellular function under stress [109,110,111].

The production of IFN-γ and TNF-α by NK cells triggers multiple immune responses. These cytokines up-regulate intercellular adhesion molecule 1 (ICAM-1) on target cells via the NF-κB signaling pathway, enhancing NK cell adhesion and the cytolytic process. IFN-γ also activates enzymes such as NOX-1,2 and nitric oxide synthase type 2 (NOS2) in macrophages. These enzymes are crucial for the oxidative burst that leads to microbial killing. NOS2, for instance, converts L-arginine into nitric oxide and citrulline, a process vital for antimicrobial defense [108]. In addition to their direct cytotoxic actions, NK cells release H_2_O_2_, which augments their migration and cytotoxic activities by affecting non-target cells like monocytes. This aspect of NK cell function is part of the indirect pathway via which they restrain infections and stimulate other immune cells, including macrophages and dendritic cells. The role of NK cells extends to the modulation of the immune response via the cytokines they release, which influence the activation and maturation of dendritic cells and the polarization of CD^4+^ T cells towards a Th1 response. This polarization is crucial for a coordinated and effective immune response against pathogens [108,112]. Moreover, the protein TXNIP within NK cells regulates IFN-γ production and is essential for proper immune functioning. TXNIP interacts with TAK1, an important kinase in TLR-induced signaling pathways that lead to cytokine production. This interaction enhances macrophage activation, phagocytosis, and the production of pro-inflammatory cytokines, thereby improving bacterial clearance [16].

## 4. Role of Thioredoxin in Pathogenic Bacterial Resistance to the Immune System

### 4.1. Role in Bacterial Response to Oxidative Stress

Bacteria, both Gram-positive and Gram-negative, have evolved complex biochemical mechanisms to cope with environmental stressors such as oxidative stress. Among these mechanisms, the thioredoxin/glutaredoxin (Trx/Grx) systems play critical roles across a broad spectrum of bacterial species, irrespective of their Gram status. These systems are instrumental in reducing oxidized cysteine residues within cellular proteins, thereby maintaining the redox equilibrium necessary for bacterial survival under stress conditions. This reduction of cysteine residues is vital for preventing the accumulation of potentially harmful oxidized proteins that could disrupt cellular functions [113].

Trx, characterized by its distinct CXXC catalytic motif, is essential for reducing oxidized cysteine residues in cellular proteins (Figure 7A). The reduction process starts when the N-terminal cysteine in the motif engages with the oxidized substrates to form a mixed disulfide intermediate. Following this, the second cysteine residue in the motif acts to complete the reduction, resulting in an oxidized Trx and a reduced substrate. This cycle of oxidation and reduction is sustained by the enzyme thioredoxin reductase, which relies on NADPH to revert Trx to its reduced state [114].

In parallel with the Trx system, Grxs also play a crucial role in reducing oxidized cysteine residues in proteins. The interaction between the Trx and Grx systems is particularly notable in *Escherichia coli*, a well-studied Gram-negative model organism. *E. coli* has two types of thioredoxins, Trx-1 and Trx-2, with Trx-1 being integral to reducing disulfide bonds in enzymes vital for cellular processes. Remarkably, *E. coli* can withstand oxidative stress even without Trx-1 and Trx-2, due to the compensatory activity of its Grx systems [114,115,116,117,118]. Mutants of *E. coli* lacking Trx-1 and Trx-2 can endure oxidative stress through the action of Grxs, which are structurally similar to Trxs and vital for redox processes in cells. Grxs are differentiated by their catalytic motifs: dithiol Grxs (like Grx1, Grx2, and Grx3) and the monothiol Grx4. Dithiol Grxs interact with and become oxidized by oxidized proteins but are restored via a two-step reaction with glutathione (GSH), producing reduced Grx and oxidized glutathione (GSSG) (Figure 7B). This GSSG is subsequently converted back to GSH by glutathione reductase, an NADPH-dependent enzyme, which completes a vital cycle of reduction and regeneration, helping maintain cellular function under oxidative stress [113,114,119]. While monothiol Grxs utilize a single cysteine residue in their active site to interact with protein disulfides, this interaction leads to the formation of a mixed disulfide bond between the cysteine of the Grx and the GSH attached to the target protein, effectively glutathionylating the protein. Subsequently, this glutathionylated intermediate is resolved by another GSH molecule, which reduces the mixed disulfide and regenerates the reduced form of the protein. Meanwhile, the Grx is returned to its reduced state, ready for another cycle. This process facilitates the reversible regulation of protein functions through modifications in their thiol groups, enhancing cellular adaptability to oxidative stress [119,120].

Given the fundamental importance of glutathione in maintaining redox equilibrium, glutathionylation emerges as an essential post-translational modification. It involves the attachment of glutathione to cysteine residues on proteins, a reversible process that protects these proteins during oxidative stress and adjusts their activity based on the cellular conditions [121,122]. Glutathionylation not only safeguards and repairs bacteria but also plays an essential role in their virulence and pathogenicity. Cellular proteins such as the death receptor Fas are crucial in responding to bacterial infections. Fas activation occurs through several pathogens, including *Pseudomonas aeruginosa*. It has been observed that Fas undergoes glutathionylation following the caspase-dependent cleavage of Grx1, which enhances the apoptotic response. Further research by the same group revealed that this modification of Fas was also present in patients suffering from *P. aeruginosa* pneumonia and facilitated the elimination of the bacteria [123,124]. Other studies have also highlighted the impact of GSH metabolism on cytokine production during *Borrelia burgdorferi* infections, which might involve glutathionylation mechanisms. Alterations in GSH-related metabolites were detected in individuals with early-stage Lyme disease, persisting for several months post-infection. Consequently, GSH metabolism and glutathionylation are proposed to be significant factors in the development of Lyme disease caused by *B. burgdorferi* and may be similarly crucial in other inflammatory conditions [125]. In Gram-positive bacteria like *Mycobacterium* and *Bacillus*, mycothiol and bacillithiol fulfill roles similar to those of glutathione in other organisms. These thiol compounds are essential not only as alternatives to glutathione but also for their ability to shield vital cellular proteins from oxidative damage. Furthermore, they are critical in controlling protein functions under stress conditions [126,127].

Moreover, glutathionylation significantly influences the function of key transcription factors in complex organisms. In the NF-κB signaling pathway, for instance, the p65 subunit undergoes glutathionylation, which impedes its DNA binding and prevents its movement to the nucleus. The enzyme Grx-1 facilitates this process by promoting deglutathionylation, thus reversing the effects of glutathionylation and ensuring normal NF-κB activation. The reduction of Grx-1 leads to decreased activation of NF-κB, affecting proteins such as IKK and TRAF-6, which illustrates Grx-1’s significant role within this pathway [128].

Trx and Grx systems are essential not only for their primary function of reducing oxidized cysteine residues via processes like glutathionylation but also for bolstering other critical antioxidant defenses. Notably, the repair of oxidized methionine (Met) residues by methionine sulfoxide reductases (Msrs) is a key process, and these Msrs rely on Trx for their regeneration, linking them closely with the Trx system and highlighting the integrated nature of the bacterial redox balance (Figure 7C) [114]. Methionine sulfoxide reductases (Msrs) specifically target methionine-S-sulfoxide (Met-S-O) with MsrA and methionine-R-sulfoxide (Met-R-O) with MsrB, necessitating the involvement of both enzymes to fully repair oxidized proteins. Despite differing in structure and sequence, MsrA and MsrB employ a common reduction mechanism. In this mechanism, a cysteine residue first attacks the oxidized methionine, triggering a cascade of reactions that form a disulfide bond and release reduced methionine. Another cysteine subsequently attacks this bond, and in some MsrA variants, an additional cysteine further facilitates enzyme recycling by altering the disulfide linkage. These processes depend on the presence of Trx, which regenerates the active form of Msr. The absence of Trx1 in certain *E. coli* mutants eliminates Msr activity, underscoring Trx’s essential role. Moreover, some eukaryotic Msrs substitute selenocysteine (Sec) for cysteine, enhancing their catalytic efficiency and suggesting an evolutionary adaptation for optimal functionality. The Trx-dependent regeneration of Msr remains a critical rate-limiting step across different Msr types [114]. This complex network of antioxidant mechanisms demonstrates the extraordinary adaptability and evolutionary sophistication of bacteria. They not only withstand but also prosper under oxidative stress, which is vital for their survival and effectiveness in harsh environments. Therefore, these systems together showcase the depth and intricacy of the strategies bacteria deploy to handle oxidative stress, underlining the essential role of Msrs in these dynamics.

### 4.2. Role in Pathogenicity and Virulence

The Trx and Grx systems are essential in maintaining the thiol–disulfide balance and defending against oxidative stress, which is critical for the survival and virulence of bacteria. Their absence significantly heightens sensitivity to oxidative agents and can prove lethal, emphasizing their essential role across various bacterial species [129,130,131]. Starting with *Streptococcus pneumoniae*, a Gram-positive bacterium, a critical reliance on extracellular thioredoxin lipoproteins, Etrx-1 and Etrx-2, along with their redox partner SpMsrAB2 was observed. These components collectively combat oxidative stress, while single mutants that maintain either one of the Etrx proteins along with SpMsrAB2 show reduced susceptibility to H_2_O_2_. In contrast, double mutants are significantly more vulnerable. This distinction highlights the proteins’ collective importance in enhancing resistance to oxidative damage and their vital role in immune evasion, particularly during pneumonia [132].

Moving to Gram-negative bacteria, *Salmonella* employs the Trx system in a central role through Trx-1, which interacts with the SsrB—a regulatory protein—to activate genes within the salmonella pathogenicity island 2 (SPI2). This interaction not only supports the bacterium’s metabolic needs but also fortifies its defenses against reactive oxygen species produced by host NADPH phagocyte oxidase, enhancing its ability to thrive in adverse host environments. The strategic use of Trx-1 highlights the adaptability and specificity of Trx systems in pathogenic contexts [133]. Extending beyond *Salmonella*, the Trx system’s versatility is further demonstrated in *Francisella tularensis*, another Gram-negative pathogen. In this bacterium, TrxA1 acts as a transcriptional regulator that modulates the expression of the oxidative stress response regulator gene (OxyR). This regulation is crucial for the bacterium’s survival and virulence as it adapts within macrophages, managing responses to oxidative stress encountered within the host [134].

The adaptability and strategic utilization of the Trx system in pathogenic bacteria such as *Francisella tularensis* highlight its crucial role in bacterial survival and virulence. This system’s importance is further illustrated in *Acinetobacter baumannii*, another Gram-negative bacterium, where TrxA is integral to modulating cell surface hydrophobicity (CSH). By maintaining type IV pili through the reduction of disulfide bonds, *A. baumannii* effectively evades macrophage capture, facilitating its interaction with host cells and enhancing its survival in the host environment [135]. Similarly, *Brucella abortus*, a Gram-negative bacterium, utilizes the Trx system to aid its pathogenicity by employing its type IV secretion system (T4SS) to suppress TXNIP expression. This suppression is critical as TXNIP is essential for the activation of NF-κB, which in turn is necessary for the production of NO and ROS within infected macrophages (Figure 8A). This suppression is a strategic adaptation that helps the bacterium modulate host cell responses and avoid immune defenses, thus promoting its persistence within the host [136].

Further expanding the scope of Trx system functionalities, *Edwardsiella piscicida*, also a Gram-negative bacterium, introduces a novel approach by using a Trx-like effector protein (Trxlp). Although structurally similar to human thioredoxin 1 (hTrx-1), Trxlp lacks redox activity. It instead interacts with key signaling molecules like IKK and Prx, obstructing the nuclear translocation of NF-κB and triggering localized accumulation of H_2_O_2_ (Figure 8B). This mechanism suppresses inflammatory responses and inhibits apoptosis signal-regulating kinase 1 (ASK-1)-driven MAPKs and cytokine production, thereby enhancing the bacterium’s ability to evade the immune system. The redox inactivity of Trxlp specifically confines NF-κB, reducing nuclear localization, which subsequently diminishes apoptosis and inflammation, further facilitating bacterial invasion [137]. These examples across various Gram-positive and Gram-negative bacteria not only illustrate the critical roles of the Trx system in managing interactions with the host but also underscore its potential as a target for developing innovative antibacterial therapies. Each bacterium’s approach to utilizing the Trx system reveals a complex panorama of pathogenic strategies, deepening our understanding of microbial adaptation and survival.

## 5. Redox-Active Therapeutic Agents in Bacterial Infection

### 5.1. Redox-Active Antibiotics

Antibiotics primarily induce bacterial cell death by inhibiting essential bacterial cell functions, classified based on the cellular component or system they affect and whether they induce cell death (bactericidal) or inhibit growth (bacteriostatic). Most current bactericidal antimicrobials target DNA, RNA, the cell wall, or protein synthesis, leading to complex alterations at biochemical, molecular, and ultrastructural levels within bacteria. The efficacy of these drugs has advanced significantly since penicillin’s discovery, driven by deeper insights into drug–target interactions and the urgent need to combat antibiotic-resistant strains. A critical aspect of how bactericidal antibiotics function involves inducing oxidative stress within bacteria. This process starts when these drugs prompt the production of hydroxyl radicals at lethal concentrations, a phenomenon stemming from alterations in the tricarboxylic acid cycle and iron metabolism [138,139].

The general mechanism through which bactericidal antibiotics induce oxidative stress within bacteria affects their metabolic pathways and essential functions. This stress often results from interference with redox processes within the bacterial cell. A key system involved in managing redox balance is the Trx system, which aids in controlling oxidative stress by facilitating the reduction of disulfide bonds in proteins. Redox-active antibiotics specifically target this system to disrupt the redox homeostasis, thereby magnifying the oxidative stress. Such targeting sets the stage for exploring more targeted interventions, including the combination of metallic ions with traditional antibiotics to enhance their effectiveness. For example, the combination of Ag^+^ with antibiotics induces ROS accumulation in *E. coli* via inhibiting electron transfer through the Trx and GSH systems, thereby increasing sensitivity to antibiotics [140,141].

Exogenous GSH supplementation decreases *E. coli* sensitivity to antibiotics through different mechanisms. The protective mechanisms of glutathione against antibiotics like streptomycin involve both simple chemical reactions and enzymatic processes. Specifically, when combined with antibiotics like ciprofloxacin, which stimulates the production of ROS in bacterial cells, glutathione plays a crucial role in mitigating oxidative stress. Through scavenging these ROS, glutathione reduces cellular damage and helps protect the bacterial cells from the oxidative effects of the antibiotic. Another mechanism utilized by glutathione, specific to streptomycin, involves the formation of a thiohemiacetal adduct between the thiol group of glutathione and the aldehyde group of the antibiotic. This chemical interaction temporarily neutralizes streptomycin, though it is a reversible reaction, requiring further metabolic modifications for the complete detoxification of the antibiotic. This indicates that glutathione acts in a sacrificial capacity to offer temporary protection. Additionally, glutathione S-transferase (GST)-mediated biotransformation presents a second protective mechanism. This enzymatic process is not exclusive to streptomycin but is also applicable to other antibiotics like ciprofloxacin and fosfomycin. In this pathway, GST enzymes catalyze the conjugation of glutathione to electrophilic sites on these antibiotics, producing glutathione conjugates. These conjugates are generally less harmful and facilitate the excretion or further metabolism of the antibiotics within bacterial systems. Significantly, the presence of glutathione transferases with high affinity towards different antibiotics has been observed in multiple bacterial systems, indicating a widespread and potentially effective defense strategy utilized by bacteria [142].

Similarly, *S. aureus*, which displays robust resistance to various antibiotics, relies heavily on the Trx system, notably TrxR, for protein disulfide reduction activity. The correlation between *S. aureus*’s susceptibility to antibiotics and oxidative stress is evident: exposure to oxidative stress upregulates transcription levels of Trx and TrxR, influencing bactericidal effects. Furthermore, oxidative stress contributes to the efficacy of antibiotics in *S. aureus*, with exogenous GSH potentially modulating susceptibility to specific antibiotics such as ciprofloxacin and gentamicin, based on oxidative stress levels [143]. These observations highlight the interconnectedness of oxidative stress management and antibiotic susceptibility in bacteria.

### 5.2. Redox-Active Sulfur/Selenium Agents

N-acetylcysteine (NAC) is a modified form of the amino acid cysteine, enhanced by the addition of an acetyl group, making it an antioxidant agent that plays a role in the oxidation and formation of disulfide bonds [18,144]. These processes are vital for stabilizing protein structures and preventing cellular damage. NAC works through several pathways: it reduces disulfide bonds in proteins, scavenges reactive oxygen species, and serves as a precursor for glutathione, a critical antioxidant in cellular defense mechanisms. In addition to its antioxidative properties [18], NAC exhibits antibacterial and anti-biofilm activities, effectively inhibiting the growth of pathogens like *S. aureus* and *E. coli* [145,146,147,148,149,150,151,152]. It competes with cysteine for uptake into bacterial cells, impairing protein synthesis and metabolic activities [18]. Its antimicrobial action is significant against both methicillin-sensitive *Staphylococcus aureus* (MSSA) and methicillin-resistant *Staphylococcus aureus* (MRSA) strains, disrupting biofilm integrity and reducing viable bacteria counts [153]. Through reducing disulfide bonds in bacterial proteins, NAC weakens essential enzymes needed for biofilm formation and alters the redox balance within cells, causing oxidative stress that damages bacterial components and leads to cell death. NAC’s ability to penetrate and destabilize bacterial biofilms also contributes to its effectiveness in inhibiting bacterial growth and potentially causing protein and DNA degradation within bacterial cells. This makes NAC a valuable adjunct to traditional antibiotics, showing potential in treating infections such as chronic bronchitis, bacteremia, and urinary tract infections [145,153,154,155].

Like NAC, cystamine (Cys) impairs the proper formation and stability of bacterial proteins by breaking down both intermolecular and intramolecular disulfide bonds. This action induces oxidative stress within bacterial cells, leading to protein dysfunction and degradation. Additionally, Cys hampers biofilm development by targeting disulfide bonds essential for the early stages of biofilm formation. It disrupts the assembly of the biofilm matrix, which consists of proteins and polysaccharides. This mechanism limits the formation and expansion of biofilms. Although its effects are somewhat less marked than those of NAC, Cys exhibits antimicrobial properties against both MSSA and MRSA strains, effectively reducing biofilm formation and decreasing the number of viable bacteria in *S. aureus* biofilms [153].

TrxR inhibition is a crucial strategy in antimicrobial therapy as it targets the enzyme essential for maintaining the redox balance in bacteria. By disrupting this balance, the growth and viability of bacteria are significantly compromised. Compounds such as auranofin, shikonin, and allicin, all of which contain sulfur, along with ebselen, a selenium-based compound, have been shown to effectively inhibit TrxR, leading to notable antimicrobial effects against Gram-positive bacteria [156]. Auranofin, a gold-containing agent approved for rheumatoid arthritis, is particularly effective against bacteria like *S. aureus*, *E. faecalis*, and *B. subtilis*, showing low minimum inhibitory concentrations and strong efficacy in various mouse models of infection, including those mimicking systemic peritonitis and topical infections, where it has been proven to combat MRSA effectively. Shikonin, a derivative from the roots of *Lithospermum erythrorhizon*, and allicin, extracted from garlic, also inhibit TrxR and display significant activity against bacteria such as *S. aureus*, *H. pylori*, and *M. tuberculosis*. In vivo studies showed that allicin can prevent biofilm formation by *S. epidermidis* in rabbit models, further underscoring the therapeutic potential of TrxR inhibitors [156]. Ebselen, characterized as a seleno–organic compound, demonstrates effective antimicrobial properties against Gram-positive bacteria, notably MRSA, *Streptococcus*, and *Enterococcus*, as well as the Gram-negative bacterium *H. pylori* [20,157]. In various mouse infection models, ebselen demonstrated effectiveness against *S. aureus*, *S. pneumoniae*, and *C. difficile* [158,159,160]. Continuing this line of research, Chen et al. synthesized and explored a series of gold (I) selenium N-heterocyclic carbene complexes and found two derivatives with remarkable antibacterial efficacy against multidrug-resistant bacteria. These derivatives disrupted bacterial TrxR, resulting in cellular DNA degradation and irreversible inhibition. In vivo assessment demonstrated substantial antibacterial activity, reduced inflammation, and accelerated recovery in mouse models. Notably, these complexes also extended survival in carbapenem-resistant *Acinetobacter baumannii* (CRAB)-induced peritonitis in BALB/c mice [161].

## 6. Future Directions

Research and development efforts in redox-active therapeutic agents for bacterial infections have primarily focused on two main avenues: redox-active antibiotics and redox-active sulfur/selenium agents. The former approach aims to understand the interactions between antibiotics and bacterial cellular functions, along with their lethal impact. However, concerns about antibiotic safety and the rise of bacterial resistance present significant challenges. On the other hand, redox-active sulfur/selenium agents primarily target host/bacterial redox systems and inhibit TrxR in bacteria. While promising, these strategies may lack broad-spectrum activity against various bacterial strains, potentially limiting their effectiveness to specific targets.

Developing targeted treatments specific to bacterial components offers promise against highly infectious and lethal bacteria. Yet, it is crucial to recognize that this approach provides only a short-term and limited solution. Therefore, there is a pressing need for more innovative solutions through the discovery and exploration of new compounds that can be used for enhancing the host immune response. Emerging as a potentially more effective strategy is the utilization of immunomodulatory peptides or TXNIP inhibitors, which could bolster the host’s innate defenses against bacterial infections. By augmenting the immune system’s ability to recognize and eradicate pathogens and defend against oxidative stress, such approaches offer a more comprehensive and sustainable solution to combat bacterial infections. This shift towards immunomodulation bridges the gap between short-term targeted treatments and long-term solutions, paving the way for more effective disease management.

### 6.1. Immunomodulatory Peptides

The exploration of immunomodulatory peptides, particularly those derived from plants, holds great promise for addressing conditions linked to compromised immune function. These peptides, diverse in their mechanisms of action, can serve as valuable candidates for drug design. An example of this potential lies in plant-derived defensins, known for their dual properties of immunomodulation and antimicrobial activity. Defensins can be instrumental as templates for the development of synthetic peptides with similar functionalities [162]. Defensins belong to diverse classes with slightly differing structures, yet they universally share characteristics as cationic peptides abundant in cysteine. The cysteine content not only supports antioxidative effects but also enables the control of reactive species production. Additionally, defensins play a pivotal role in modulating both innate and adaptive immune responses by activating macrophages and neutrophils and participating in signal transduction pathways [162]. However, the dynamics between these peptides and their impact on the thioredoxin system remain underexplored.

Additional examples of immune modulatory peptides include cryptic peptides, which exhibit structural heterogeneity. These peptides have been confirmed to be generated from plant proteins in response to antigens. They play a role in influencing innate immunity by activating natural killer cells. A peptide referred to as GmSubPep was isolated. Originating from an extracellular subtilisin-like protease (subtilase), GmSubPep has the capability to initiate the MAPK signaling cascade by binding to putative receptors in membranes. Furthermore, a peptide named CAP-derived peptide 1 (CAPE1) has been demonstrated to enhance the transcription of genes associated with antioxidative defense and modulate protein–protein interactions [163,164,165]. The innovative design of immunomodulatory peptides targeting human thioredoxin presents a promising therapeutic avenue. The aim is to engineer peptides that either bind to the active site of thioredoxin or influence its interactions with other proteins, enhancing or mimicking the redox capabilities of thioredoxin. This could potentially create a more favorable cellular reducing environment and bolster the immune system. These peptides may achieve this by mimicking thioredoxin’s actions or by regulating its activity, which could increase its reducing capacity and impact the cellular redox environment. However, the development of these peptides comes with challenges, including potential side effects, scalability of production, and the complexities involved in designing peptides that precisely target specific components of the immune system.

Similarly, cryptic peptides derived from larger plant proteins have considerable therapeutic potential for regulating analogous pathways in humans, particularly in boosting immune responses and managing oxidative stress. These peptides can enhance the transcription of genes associated with antioxidative defenses and modulate protein–protein interactions, thereby playing a role in activating innate immunity, including natural killer cells, and influencing the MAPK signaling cascade through binding to specific receptors on cell membranes. Like the peptides that target thioredoxin, cryptic peptides require synthesis or exogenous production for use in human therapies.

Crafting peptides inspired by defensin to mimic the action of human thioredoxin involves utilizing defensins’ structural features that can be tailored to perform similar redox operations. This strategy encompasses selecting and modifying defensin peptides to incorporate thioredoxin’s active site or emulate its electron transfer capabilities, thereby adjusting cellular redox balances. Similarly, the production of cryptic peptides that target biochemical pathways associated with oxidative stress and immune response follows the same synthesis methodology. Both types of peptides can be synthesized using recombinant DNA technology [166,167], which involves genetically engineering bacteria to produce these peptides in a scalable and precise manner. This approach combines bioinformatics [168], molecular modeling [169], solid-phase peptide synthesis [170], and functional assays to ensure the peptides’ efficacy, stability, and biological suitability. The method includes ongoing improvements, using insights from functional assays to optimize the peptide design. This ensures that the peptides maintain their structural integrity and biological suitability while enhancing their effectiveness. Through this controlled synthesis process, researchers can address the challenges associated with peptide development and create effective treatments for diseases linked to oxidative stress and immune dysfunction.

### 6.2. TXNIP Inhibitors

Another interesting idea is the development of TXNIP inhibitors, which represents a promising therapeutic strategy for enhancing immunity against pathogenic bacterial infections, increasing the antioxidant activity of Trx, and reducing apoptosis. Research has demonstrated that TXNIP plays a critical role in modulating the immune response during bacterial infections. Through inhibiting TXNIP, there is a significant enhancement in NK cell-mediated innate immunity, marked by increased IFN-γ production. This upregulation of IFN-γ is pivotal for activating macrophages, in turn, boosting the immune system’s ability to clear bacterial infections efficiently. This strategy suggests not only a method for enhancing the body’s defense against infections but also a way to maintain a robust immune cell population, essential for continuous protection against pathogens. However, the development of TXNIP inhibitors will require careful consideration of the broader regulatory roles of TXNIP in cellular processes, including metabolism, inflammation, and cell cycle regulation, to mitigate unintended consequences of systemic TXNIP inhibition [16,171,172].

The exploration and development of inhibitors targeting TXNIP have marked significant advancements in the field of medical research, particularly in the context of chronic and degenerative diseases. Various small-molecule compounds and phytochemicals have been identified and studied for their capacity to modulate TXNIP expression and its associated signaling pathways. For instance, verapamil, an anti-hypertensive medication, has been shown to inhibit TXNIP transcription in models of diabetes and Alzheimer’s disease, thus preventing β-cell apoptosis and enhancing glucose homeostasis. Similarly, metformin, a frontline medication for type 2 diabetes, effectively prevents the activation of the TXNIP/NLRP-3 inflammasome by mitigating ER stress and activating AMPK in macrophages. Other notable inhibitors such as SRI-37330, DI-NBP, W2476, quercetin, allopurinol, and salidroside have demonstrated promising effects across various disease models, including diabetes, nonalcoholic fatty liver disease, and diabetic nephropathy, targeting different mechanisms of TXNIP action [173].

Despite these advances, investigation into the potential of TXNIP inhibitors to enhance the immune system in the context of bacterial infections remains relatively incomplete. The existing body of research primarily focuses on chronic diseases such as diabetes, neurodegenerative disorders, and conditions characterized by excessive inflammation or oxidative stress. The multifunctional nature of TXNIP and its central role in disease pathogenesis suggests that inhibiting this protein could offer novel therapeutic avenues not only for these conditions but also for modulating the immune response to bacterial infections. This area represents a promising frontier for future research, which could uncover new strategies to bolster the immune defense against pathogens through leveraging the regulatory mechanisms of TXNIP. Identifying and developing effective TXNIP inhibitors for this purpose would not only broaden the understanding of TXNIP’s role in immunity but also potentially lead to innovative treatments for infectious diseases.

## 7. Conclusions

In conclusion, the interplay between host organisms and microbial pathogens entails a dynamic immune response characterized by complex signaling pathways, including NF-κB and MAPK, as well as the activation of PRRs like TLRs and NLRs. Redox systems, including Trx/Grx, are essential in maintaining intracellular redox balance and regulating immune cell functions. On the other hand, bacteria employ multiple resistance strategies to avoid the immune system, including modulation of enzymatic antioxidants, redox-regulated adaptation, iron regulation, and modulation of the host immune system. Redox-active antibiotics and sulfur/selenium agents show promising potential for therapeutic interventions, although they are currently under study and concerns have been raised, particularly in the context of antibiotic use. Exploring the host immune–redox interaction in bacterial infections may provide a potential approach for therapeutic strategies.

## Figures and Tables

**Figure 1 antioxidants-13-00545-f001:**
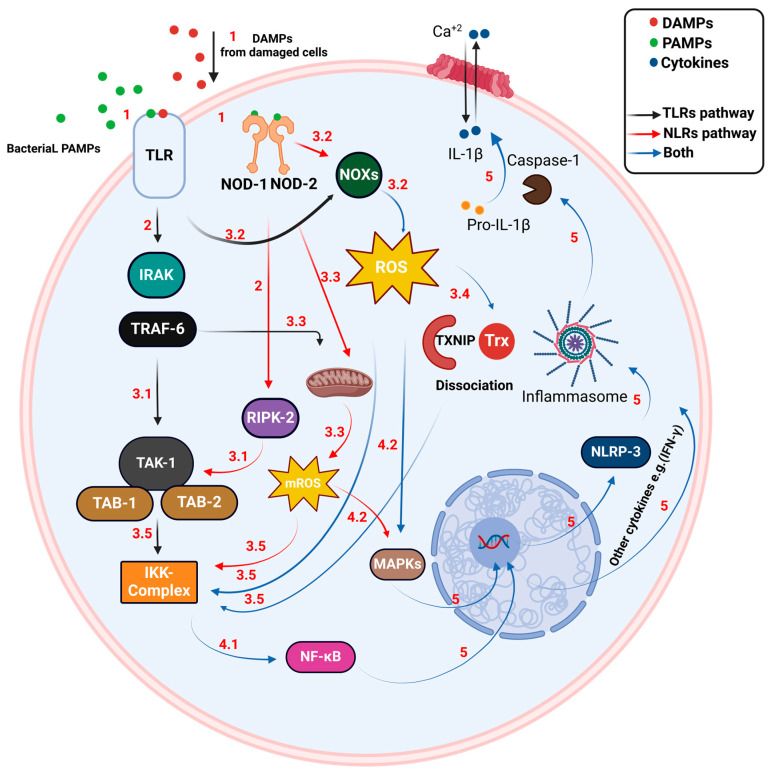
Overview of TLR and NOD receptor-mediated immune response. Step 1: Pathogen Detection: TLRs identify extracellular/endosomal PAMPs and DAMPs, such as lipopolysaccharides and heat shock proteins. NOD-1 and NOD-2 detect intracellular peptidoglycan fragments. Step 2: Signal Transduction: TLRs activate signaling leading to kinase activation (IRAKs, TRAF-6). NOD receptors directly activate RIPK-2 kinase. Step 3: Activation of Signaling Complexes, ROS Production, and Response Amplification: 3.1. Activation of Signaling Complexes: both receptor types stimulate the TAK-1/TAB complex, which then activates the IKK complex essential for NF-κB pathway initiation. 3.2. Activation of NOX: Triggered by NOD-2 signaling or TLRs, NOX generates reactive oxygen species (ROS). This directly activates the IKK complex. 3.3. Role of TRAF-6 and NOD-2: TRAF-6 and NOD-2 can enhance the production of mitochondrial ROS (mROS), which further activates the IKK complex. 3.4. Dissociation of Trx from TXNIP: the increase in ROS causes Trx to dissociate from TXNIP. 3.5. Activation of the IKK Complex: each of these steps synergistically contributes to the activation of the IKK complex, central to regulating inflammatory responses and immune function. Step 4: NF-κB-Mediated Transcription and MAPK Activation: 4.1. NF-κB-Mediated Transcription: IKK complex phosphorylates IκB, allowing NF-κB to enter the nucleus and upregulate inflammatory genes, including cytokines and chemokines. 4.2. MAPK Activation: ROS and mROS contribute to the activation of the MAPK pathway. Step 5: Inflammatory Response and Immune Activation: NF-κB facilitates the assembly of the NLRP-3 inflammasome, which promotes the production of cytokines and chemokines. Concurrently, the activation of the MAPK pathway leads to the release of cytokines and interferons (IFN), thereby driving the recruitment of immune cells such as macrophages and neutrophils. This cascade enhances inflammation and pathogen neutralization through phagocytosis and the subsequent production of ROS/RNS. (PAMPs: pathogen-associated molecular patterns; PRRs: pattern recognition receptors; TLRs: toll-like receptors; NLRs: nucleotide-binding oligomerization domain-like receptors; DAMPs: damage-associated molecular patterns; IRAK: IL-1 receptor-associated kinase; TRAF-6: TNF receptor-associated factor 6; TAK-1: TGF-beta-activated kinase 1; TAB: TAK1-binding protein; IKK: IκB kinase; IκB: inhibitor kappa B; NF-κB: nuclear factor kappa-light-chain-enhancer of activated B cells; mROS: mitochondrial reactive oxygen species; NOD-1/2: nucleotide-binding oligomerization domain 1/2; RIPK-2: receptor-interacting protein kinase 2; NLRP-3: NLR family pyrin domain containing 3; IL-1β: interleukin-1 beta; NOX: NADPH oxidase; ROS: reactive oxygen species; MAPK: mitogen-activated protein kinase; Trx: thioredoxin; TXNIP: thioredoxin-interacting protein; TNF-α: tumor necrosis factor-alpha; ILs: interleukins; IFNs: interferons).

**Figure 2 antioxidants-13-00545-f002:**
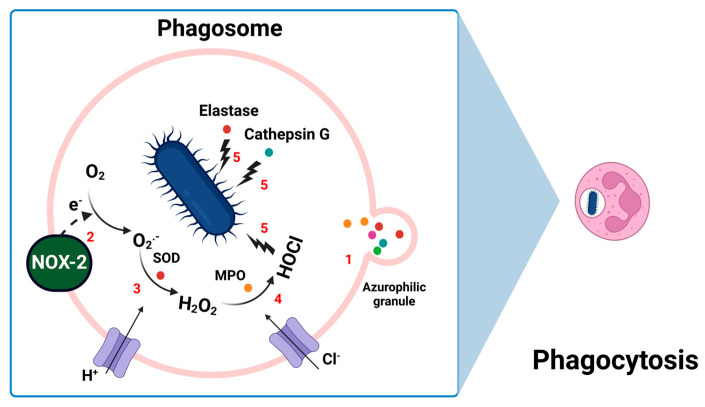
Phagocytosis in neutrophils. Step 1: Mobilization of Azurophilic Granules: Neutrophils mobilize azurophilic granules, specialized lysosomes containing antimicrobial proteins such as MPO, elastase, and cathepsin. The azurophilic granules fuse with the phagosome, releasing their contents to interact with the engulfed pathogen. Step 2: Activation of NOX: the NOX-2 complex on the phagosomal membrane becomes activated, catalyzing the transfer of electrons from NADPH to molecular oxygen, generating ROS in the form of O_2_^−^. Step 3: Conversion of Superoxide to Hydrogen Peroxide: SOD within the phagosome catalyzes the conversion of superoxide anion into H_2_O_2_. Step 4: Myeloperoxidase-Catalyzed Reactions: MPO released from azurophilic granules uses hydrogen peroxide as a substrate to form HOCl, a potent antimicrobial agent. Step 5: Synergistic Enzymatic Action: enzymes like elastase and cathepsin, alongside HOCl, degrade bacterial proteins, nucleic acids, and cell walls, effectively destroying the pathogen. (MPO: myeloperoxidase; NOX-2: NADPH oxidase 2; ROS: reactive oxygen species; O_2_^−^: superoxide anion; SOD: superoxide dismutase: H_2_O_2_: hydrogen peroxide; HOCl: hypochlorous acid).

**Figure 3 antioxidants-13-00545-f003:**
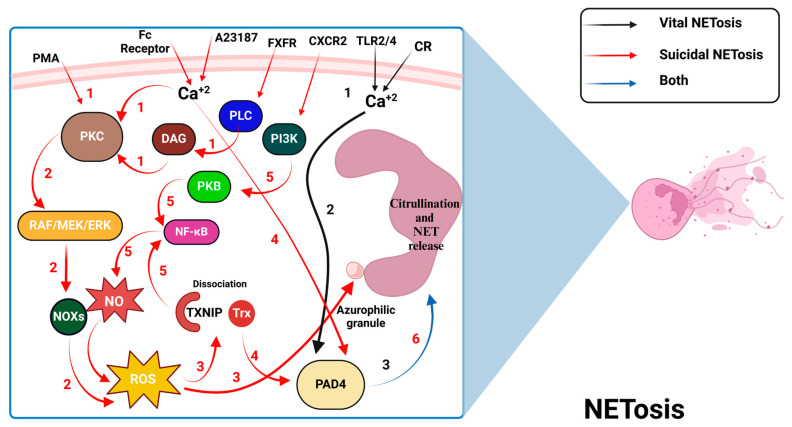
NETosis in neutrophils. (Suicidal NETosis). Step 1: Activation of PKC-α: Various agents such as PMA directly stimulate PKC-α. Additionally, increased intracellular Ca^2^⁺ due to antibodies and the calcium ionophore A23187 elevate intracellular calcium levels, subsequently activating PKC-α. FGFR activation triggers PLC-γ, producing DAG to enhance PKC-α activity. Step 2: Cascade of Biochemical Reactions: PKC-α initiates the RAF kinase/threonine protein kinase/mitogen-activated protein kinase/extracellular signal-regulated kinase (RAF/MEK/ERK) signaling pathway, which then stimulates NOX, leading to ROS production. Step 3: ROS-Mediated Dissociation and Granule Release: Elevated ROS levels cause the oxidative dissociation of Trx from TXNIP, releasing Trx. Furthermore, ROS facilitates the release of azurophilic granules containing MPO, elastase, and cathepsin, which are crucial for the nuclear processes involved in suicidal NETosis. Step 4: Activation of PAD4: Increased intracellular calcium, due to the calcium ionophore A23187 and antibodies, can directly activate PAD4. Furthermore, freed Trx offers an alternative activation route for PAD4, distinctively without utilizing its oxidoreductive function. Step 5: Inflammatory Signaling Loop: CXCR activation by IL-8 initiates the PI3K/PKB pathway, promoting downstream signaling that activates NF-κB. NF-κB activation, stimulated by the PI3K/PKB pathway and TXNIP after dissociation from TRX, enhances iNOS production. This is essential for NO generation, which exacerbates inflammation and reinforces the suicidal NETosis feedback loop. Step 6: Histone Citrullination and NET Formation: PAD4 converts arginine to citrulline in histones, leading to histone citrullination and promoting NET formation, essential for both types of NETosis. (Vital NETosis). Step 1: Bacterial Induction and Calcium Elevation Pathway: Bacteria like *E. coli* and *S. aureus*, or bacterial-specific molecular patterns recognized by PRRs, induce vital NETosis. Activation of receptors such as TLR-2, TLR-4, and CR raises intracellular Ca^2^⁺ levels, facilitating PAD4 activation. Step 2: Activation of PAD4 and Histone Modification: similar to suicidal NETosis, elevated Ca^2^⁺ levels activate PAD4, which then mediates histone citrullination. Step 3: Histone Decondensation and NET Release: histone citrullination causes decondensation, and in vital NETosis, NETs are released from vesicles without causing membrane rupture or neutrophil death. (NETosis: neutrophil extracellular trap formation; PKC-α: protein kinase C alpha; PMA: phorbol myristate acetate; FGFR: fibroblast growth factor receptor; PLC-γ: phospholipase C-gamma; DAG: diacylglycerol; RAF/MEK/ERK—RAF kinase/mitogen-activated protein kinase/extracellular signal-regulated kinase signaling pathway; NOX: NADPH oxidase; ROS: reactive oxygen species; Trx: thioredoxin; TXNIP: thioredoxin interacting protein; MPO: myeloperoxidase; PAD-4: peptidylarginine deiminase 4; CXCR: C-X-C motif chemokine receptor; IL-8: interleukin 8; PI3K/PKB: phosphoinositide 3-kinases/protein kinase B signaling pathway; NF-κB: nuclear factor kappa-light-chain-enhancer of activated B cells; iNOS: inducible nitric oxide synthase; NO: nitric oxide; PRRs: pattern recognition receptors; TLR-2, TLR-4: toll-like receptor 2, toll-like receptor 4; CR: complement receptor).

**Figure 4 antioxidants-13-00545-f004:**
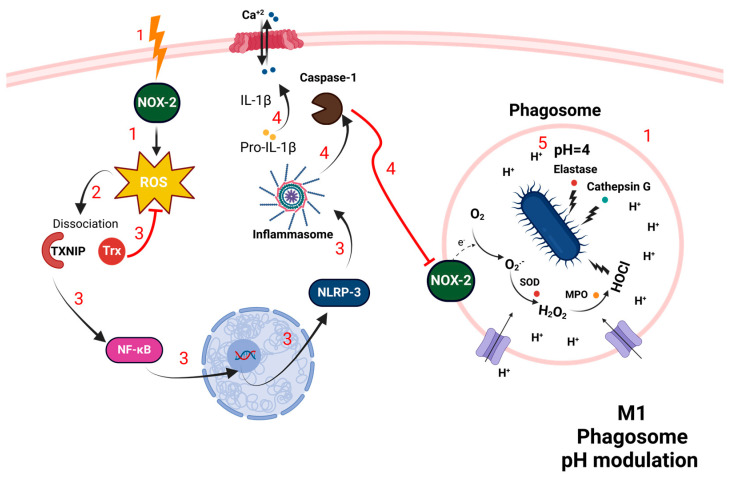
M1 macrophage pH regulation. Step 1: Bacterial Engulfment and NOX-2 Activation: macrophages engulf bacteria, activating NOX-2, which produces ROS. Step 2: TXNIP Dissociates from Trx: ROS triggers the dissociation of TXNIP from Trx, freeing TXNIP. Step 3: Activation of NF-κB and NLRP-3: Free TXNIP activates the transcription factor NF-κB and the NLRP-3 inflammasome, crucial for inflammatory responses. Concurrently, free Trx inhibits ROS. Step 4: Caspase-1 Activation: NLRP-3 activates caspase-1, which processes inflammatory cytokines and regulates ROS levels, maintaining an acidic phagosomal environment (pH 4). Step 5: Enhanced Bacterial Degradation: the acidic pH activates enzymes that degrade bacteria, promoting effective immune responses [17,24,29,57]. (NOX-2: NADPH oxidase 2; ROS: reactive oxygen species; TXNIP: thioredoxin-interacting protein; Trx: thioredoxin; NF-κB: nuclear factor kappa-light-chain-enhancer of activated B cells; NLRP-3: NLR family pyrin domain containing 3; IL-1β: interleukin 1 beta).

**Figure 5 antioxidants-13-00545-f005:**
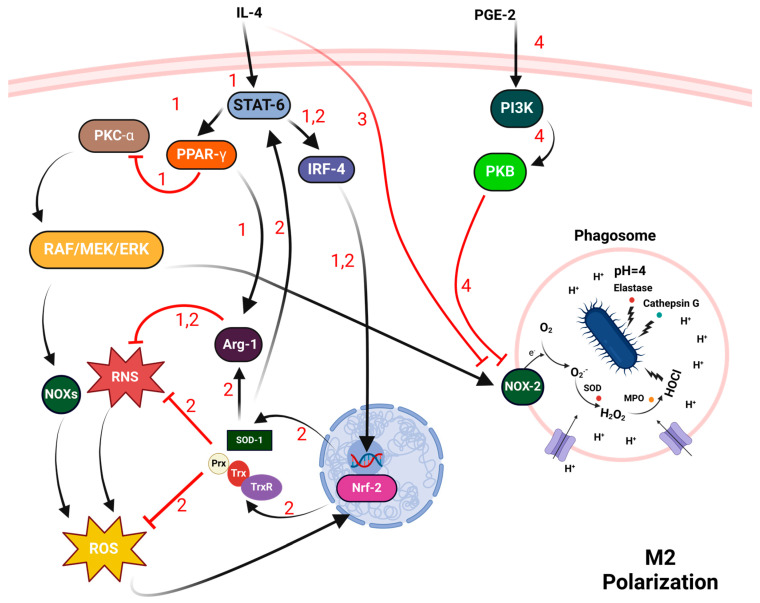
M2 macrophage polarization. Step 1: Activation of STAT-6/PPAR-γ by IL-4: IL-4 stimulates PPARγ activation through STAT-6, which inhibits PKC-α and diminishes NOX-mediated ROS generation via the RAF/MEK/ERK pathway. Furthermore, the activation of STAT-6 and IRF-4 produces H_2_O_2_, which is essential for starting the transcription of M2-specific genes. IL-4 increases Arg-1 activity, shifting metabolism from nitric oxide (via iNOS) to ornithine, supporting tissue repair, via STAT-6/PPAR-γ [63,64]. Step 2: Nrf-2 Activation and Oxidative Stress Defense Enhancement: Nrf-2 regulates expression of SOD-1 and Trx systems by binding to antioxidant response elements (AREs), enhancing defense against oxidative stress and promoting M2 gene transcription [63,64]. SOD-1 assists in the activation of STAT-6 and IRF-4 by producing H_2_O_2_, which is essential for starting the transcription of M2-specific genes. Additionally, the overexpression of SOD-1 leads to a decrease in iNOS gene expression and a reduction in NO synthesis, while simultaneously promoting arginase I expression and increasing urea production These alterations affect collagen synthesis and play a role in the development of fibrosis [65,66]. The Trx system actively inhibits ROS production, whereas the Prx system specifically inhibits NO production. This targeted inhibition is crucial for reducing oxidative stress and supporting cellular repair mechanisms, thereby promoting M2 macrophage polarization [67,68]. Step 3: NOX-2 Downregulation and Lysosomal Enhancement: IL-4 directly downregulates NOX-2 and enhances expression of cathepsins S and L, improving phagosome protein degradation efficiency [62]. Step 4: Activation of PI3K/PKBPathway by PGE-2: PGE-2 activates the PI3K/PKB pathway, further suppressing NOX-2 activation and aiding M2 polarization [69]. (M2: M2 type macrophage; STAT-6: signal transducer and activator of transcription 6; PPAR-γ: peroxisome proliferator-activated receptor gamma; IL-4: interleukin 4; PKC-α: protein kinase C alpha; NOX: NADPH oxidase; ROS: reactive oxygen species: RAF: RAF kinase; MEK: mitogen-activated protein kinase; ERK: extracellular signal-regulated kinase; IRF-4: interferon regulatory factor 4; H_2_O_2_: hydrogen peroxide; Arg-1: arginase 1; iNOS: inducible nitric oxide synthase; Nrf-2: nuclear factor erythroid 2-related factor 2; SOD-1: superoxide dismutase 1; Trx: thioredoxin; AREs: antioxidant response elements; Prx: peroxiredoxin; NOX-2: NADPH oxidase 2; PI3K: phosphoinositide 3-kinase; PKB: protein kinase B; PGE-2: prostaglandin E2).

**Figure 6 antioxidants-13-00545-f006:**
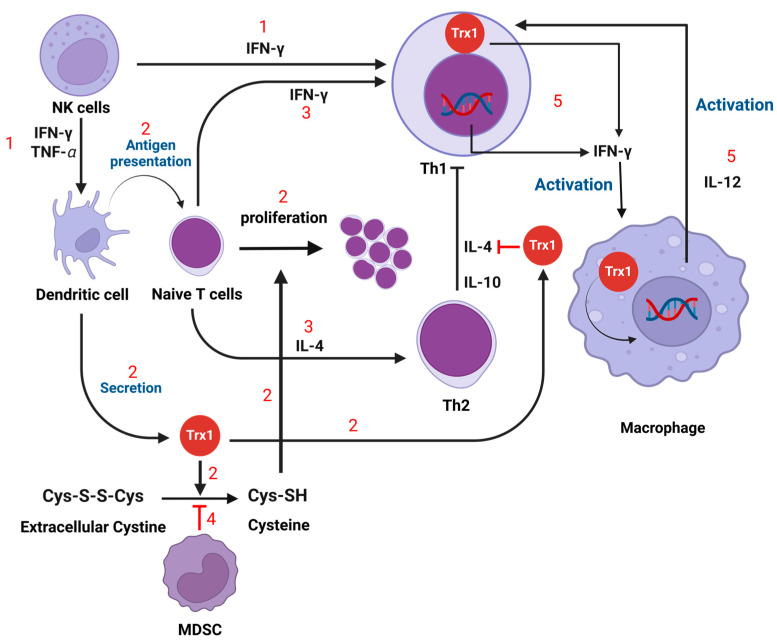
Trx1 role in proliferation and differentiation of T cells. Step 1: NK Cells’ Role: NK cells influence the maturation of dendritic cells by releasing TNF-α and IFN-γ. Not only does IFN-γ mature DCs, but it also significantly drives the differentiation of Th1 cells. Step 2: DCs’ Role: Mature dendritic cells capture antigens and present them to naive T cells. This antigen presentation is crucial for the initial activation of the naive T cells, setting the stage for the adaptive immune response. The protein Trx1, secreted by dendritic cells, regulates the expression of CD4 and CD30 receptors on T cells. Trx1 not only adjusts receptor expression but also facilitates the conversion of cystine to cysteine, crucial for T-cell proliferation. Trx-1 deactivates IL-4, suppressing the Th2 immune response and favoring Th1. Step 3: Influence of Cytokines on Differentiation: Naive T cells differentiate into various helper T-cell subsets under the influence of local cytokines. IL-4 drives the differentiation into Th2 cells, while IFN-γ promotes the development of Th1 cells. Step 4: MDSCs’ Role: MDSCs limit cystine availability by competing for this molecule, which is essential for its conversion by antigen-presenting cells. This competition impairs T-cell function and promotes immune suppression by interacting with Trx1. Step 5: Feedback Loop between Th1 Cells and Macrophages Involving Trx-1 and IFN-γ: Trx1 enhances IFN-γ production in Th1 cells, which in turn promotes higher levels of Trx1. This feedback loop also involves IFN-γ-activated Trx1 in macrophages, which boosts IL-12 secretion by modulating the thiol redox state. (Trx1: thioredoxin 1; NK cells: natural killer cells; TNF-α: tumor necrosis factor alpha; IFN-γ: interferon gamma; DCs: dendritic cells; T cells: T lymphocytes; CD4: cluster of differentiation 4; CD30: cluster of differentiation 30; IL-4: interleukin 4; Th1: T helper type 1 cells; Th2: T helper type 2 cells; MDSCs: myeloid-derived suppressor cells; IL-12: interleukin 12).

**Figure 7 antioxidants-13-00545-f007:**
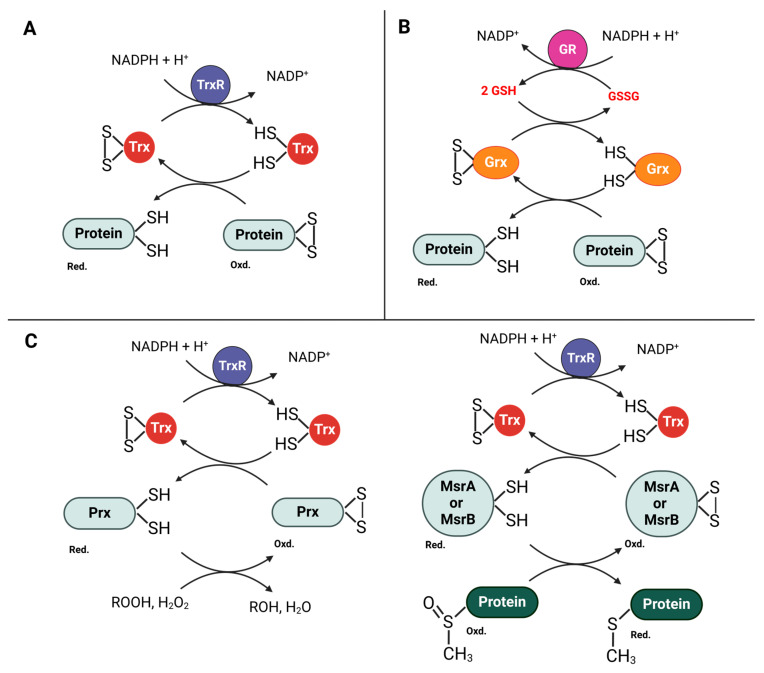
Role of Trx and Grx in oxidative stress response in bacteria. (**A**) The Trx system begins with the reduction of oxidized cysteine residues in proteins, facilitated by the CXXC motif in Trx. This involves an initial attack by the N-terminal cysteine of Trx to form a mixed disulfide with the substrate, followed by the action of the second cysteine to release a reduced substrate and oxidized Trx. The oxidized Trx is then regenerated by thioredoxin reductase, using NADPH as a reducing agent. (**B**) The process starts when dithiol Grx, using its N-terminal cysteine, attacks a disulfide bond in a substrate protein, creating a mixed disulfide complex. Then, the C-terminal cysteine of Grx also attacks, releasing a reduced substrate and turning Grx into its oxidized form. This oxidized Grx is then returned to its reduced state by two GSH molecules, forming GSSG in the process. Finally, GR converts the GSSG back into GSH. (**C**) Trx provides the electrons to peroxiredoxin and Msr in the defence against oxidative stress. Prx scavenges free radicals directly, and Msrs reduces free oxidized methionine or oxidized methionine residues in proteins. Trx is essential for regenerating active Msrs, highlighting its indispensable role across different systems. (Trx: thioredoxin; TrxR: thioredoxin reductase; Grx: glutaredoxin; GR: glutathione reductase; GSH: glutathione; GSSG: glutathione disulfide; Prx: peroxiredoxin; Met: methionine; MsrA: methionine sulfoxide reductase A; MsrB: methionine sulfoxide reductase B; Oxd.: oxidized; Red.: reduced).

**Figure 8 antioxidants-13-00545-f008:**
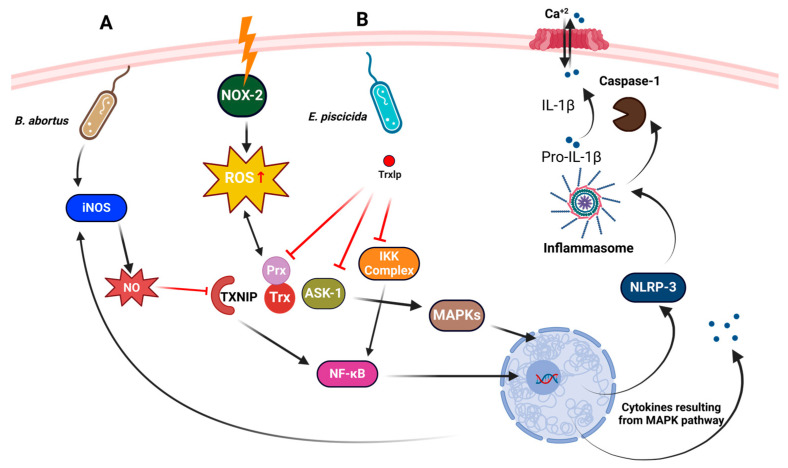
Mechanisms of immune suppression during bacterial infection. (**A**) Infection with *B. abortus* causes the expression of iNOS and production of NO, which suppress the expression of TXNIP within infected macrophages. Since TXNIP is crucial for activating NF-κB, its suppression leads to decreased activity of this critical transcription factor. With NF-κB activity diminished, the production of NO and ROS, both vital for immune defense, is also reduced. (**B**) *Edwardsiella piscicida* secretes Trxlp, a protein that mimics the structure of human hTrx1 but is devoid of redox activity. Trxlp binds directly to the IKK complex, which is essential for the activation of the NF-κB signaling pathway. This pathway is pivotal for initiating immune responses such as inflammation and apoptosis. The direct engagement of Trxlp with IKK obstructs the nuclear translocation of NF-κB. Without NF-κB in the nucleus, the transcription of immune response genes is hindered, which in turn reduces inflammation and apoptosis. Trxlp also interacts with Prx enzymes that normally reduce H_2_O_2_. This interaction instead leads to a localized accumulation of H_2_O_2_. The increase in H_2_O_2_, facilitated by Trxlp’s interaction with Prx, suppresses apoptosis signal-regulating kinase 1 (ASK1). ASK1 is involved in activating MAP kinases, key players in stress response pathways including those governing inflammation and cell death. Suppressing ASK1 leads to reduction in the activation of MAP kinases and the production of pro-inflammatory cytokines. This results in a weakened immune response, creating a more favorable environment for bacterial survival. (TXNIP: thioredoxin-interacting protein; NF-κB: nuclear factor kappa-light-chain-enhancer of activated B cells; NO: nitric oxide; ROS: reactive oxygen species; Trxlp: thioredoxin-like protein; hTrx1: human thioredoxin 1; IKK: IκB kinase inhibitor of kappa B; Prx: peroxiredoxin; H_2_O_2_: hydrogen peroxide; ASK1: apoptosis signal-regulating kinase 1; MAPKs: mitogen-activated protein kinases).

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
