# Peer review of "Exploring Immune Redox Modulation in Bacterial Infections: Insights into Thioredoxin-Mediated Interactions and Implications for Understanding Host–Pathogen Dynamics"

_antioxidants, 2024, doi:10.3390/antiox13050545_

Round 1
Reviewer 1 Report
In this review the authors deal with the intricate network of host-pathogen interactions, focusing on redox mechanisms and thioredoxin system role in immune response to bacterial infections.
This is an interesting research area because -as the authors highlight- could pave the way for new intervention strategies in infectious diseases.
Revision of some points (detailed below) could further improve the work.
Revision of the following points are necessary to improve the work:
· Figure legends.
In the legend to Figure 3 it is described the late stage of macrophage phagosome maturation; it should contain the description of ROS/Trx pathway too (the left part of the Fig. 3A).
Similarly, the legend to Figure 4 should more precisely describe what the figure shows.
In the Figure 6 I cannot see S. typhimurium inhibition of apoptosis.
· Redox-Active Antibiotics (paragraph 5.1)
It should be clarified how supplementation of exogenous GSH, known as antioxidant, can modulate susceptibility to antibiotics (Lanes 590-592, 596-599).
· Paragraph 5.2
Specify that among Gram negative there is H. pylori (Lane 641)
Desirable:
"Parallel to the Trx system, Grxs contribute significantly to the reduction of oxidized cysteine in proteins” (lanes 474 and following) that can also to be linked to glutathione cysteine, in the process of glutathionylation/deglutathionylation. As glutathionylated proteins are shown to be involved in infection and inflammation, it could be of interest to add a comment on them in the paragraph and figure 5B, or alternately, in the introduction.
Author Response
In this review the authors deal with the intricate network of host-pathogen interactions, focusing on redox mechanisms and thioredoxin system role in immune response to bacterial infections.
This is an interesting research area because -as the authors highlight- could pave the way for new intervention strategies in infectious diseases.
>>> We sincerely thank you for recognizing the importance of our review's focus on the intricate network of host-pathogen interactions and the role of redox mechanisms and the thioredoxin system in the immune response to bacterial infections. We appreciate your acknowledgment of the significance of this research area and its potential to lead to new intervention strategies in infectious diseases. Your insights and constructive feedback are invaluable to us as we strive to enhance the clarity and impact of our work.
Revision of some points (detailed below) could further improve the work.
Detail comments
Revision of the following points are necessary to improve the work:
- Figure legends.
In the legend to Figure 3 it is described the late stage of macrophage phagosome maturation; it should contain the description of ROS/Trx pathway too (the left part of the Fig. 3A).
Figure 3: In response to the feedback, we have not only revised the legend for Figure 3 but also divided the original Figure 3 into multiple figures to enhance clarity and focus. The revised Figure 3 now specifically includes a detailed depiction of the ROS/Trx pathway as it relates to macrophage phagosome maturation, with its legend updated accordingly to accurately reflect this focus. This separation allows each aspect of the original figure to be explored in greater detail and improves the overall readability and effectiveness of the visual presentation.
Similarly, the legend to Figure 4 should more precisely describe what the figure shows.
>>>Figure 4 now renumbered as Figure 6: Following our recent restructuring of the figures to improve clarity, what was previously referred to as Figure 4 has now been renumbered as Figure 6. The legend for this figure has been extensively updated to more precisely describe the contents and focus depicted in the new Figure 6, ensuring its clarity and relevance to the accompanying text. This renumbering reflects the addition and division of earlier figures to better detail and separate the discussions within our review.
In the Figure 6 I cannot see S. typhimurium inhibition of apoptosis.
>>>Figure 6 now renumbered as Figure 8: Upon reviewing your comments regarding the absence of S. typhimurium inhibition of apoptosis in Figure 8, we realized that this was due to an oversight in the final editing of the figure legends. The depiction and description of S. typhimurium inhibition of apoptosis were indeed part of our initial drafts but were mistakenly not removed from the text during the final revisions. We have now corrected this error, ensuring that the legend accurately reflects the contents of the figure and aligns with the discussions in the text.
Redox-Active Antibiotics (paragraph 5.1)
It should be clarified how supplementation of exogenous GSH, known as antioxidant, can modulate susceptibility to antibiotics (Lanes 590-592, 596-599).
>>>We have clarified how supplementation with exogenous GSH, as an antioxidant, can modulate susceptibility to antibiotics, expanding on the mechanisms involved in lines 590-592 and 596-599.
Paragraph 5.2
Specify that among Gram negative there is H. pylori (Lane 641)
>>>We have specified that among Gram-negative bacteria, H. pylori is included, as noted in line 641.
Desirable:
"Parallel to the Trx system, Grxs contribute significantly to the reduction of oxidized cysteine in proteins” (lanes 474 and following) that can also to be linked to glutathione cysteine, in the process of glutathionylation/deglutathionylation. As glutathionylated proteins are shown to be involved in infection and inflammation, it could be of interest to add a comment on them in the paragraph and figure 5B, or alternately, in the introduction.
>>>We have included a comment on glutathionylated proteins and their role in infection and inflammation in paragraph 5.2, enriching the discussion on redox regulation during bacterial infections.
Reviewer 2 Report
In the review entitled “Exploring Immune Redox Modulation in Bacterial Infections: Insights into Thioredoxin-Mediate Interactions and Implications for Understanding Host-Pathogen Dynamics” Dagah et al. discuss the function of the Trx system in both immune cells and bacterial cells and how it can be modulated during infection to the host’s advantage. While the content of this review is very interesting, there are several issues I have with the paper.
Major issues:
· In sections 2 and 3, when talking about pathways, the authors should discuss it in a linear fashion, stating with initiators, receptors, signaling intermediates, transcription factors, genes expressed. They frequently start somewhere in the middle and then might talk about initiators then transcription factors, then something else in the middle. Discussing them in a linear fashion allows the reader to follow along like you are telling a story.
o In Lines 211-231, you back track. You talk about suicidal NETosis, then vital, then back to suicidal. Talk about suicidal then vital, then you can talk about commonalities or differences.
o Lines 290-305, when describing the pathways, the order of molecules is very confusing and many of the molecules are not listed in Fig. 3B where the pathways is supposed to be illustrated.
· There is frequently awkward wording where a word is not really the best choice.
o Line26 – “exhibited their relevance.”
o Line 618 – adjunct
o Line 719 - conceive
· Figures are often confusing, and the legend is not adequate.
o Figure 1 – this is too many pathways in one figure, the reader can’t keep up with it. To keep it in one figure, I would use color coordination. Initiator one color, receptors another color, intermediates a color, transcription factor get a color, etc. Then I would number the steps – you can use colored arrows to denote separate pathways.
§ PAMPs are not necessarily secreted from the bacteria so this drawing can be misleading.
§ Why are some arrows dotted lines?
§ Cytokines show be shown secreted from the cell.
§ ASC is part of the inflammasome not downstream.
o Figure 2: Direction in the figure would be helpful.
§ Numbered steps in both A and B
§ Move A and B to the top of the cell.
§ Love the coloring of the arrows but why are some solid and some dotted?
§ Line 190, O2 has a weird red line after it.
o Figure 3, Direct the reader using numbers. Also, the legends can’t just start at step 8 (caspase 1), you have to explain the whole figure.
§ Panel B- arrow to Arg1 should be shortened to end before the protein line other arrows.
§ Why are some arrows dotted lines?
§ In the legend for panel B, the first sentence is not a sentence.
o Figure 4 needs updated.
§ I think there are several locations where the text describes Trx1 inhibition that are not illustrated. Specifically, with going from naïve T cells to Th1 or Th2 cells.
§ There is a macrophage that is not discussed in the legend.
§ In the legend, Sentence 2 is not a sentence – was this maybe an alternative title for the figure? Maybe just delete it.
o Figure 5 needs update.
§ All arrows that say reduction are actually a oxidation reaction because the product is the oxidized protein.
§ In Panel B, should MsrA/B be attached to met instead of cys?
§ In abbreviations, can you match the abbreviation to the one in the diagram?
· Ex Grx- thiol or disulfide bond I think are GSH and GSSG (could also make then unique colors that are define with the abbreviation)
o Figure 6 needs updating
§ T4SS is embedded in the bacteria is a secretion product you are referring to?
§ Most of panel B is not described in the legend. Describe it or remove it from the figure.
§ Legend talks about S. typhimurium but the figure does show that.
§ Legend does not need to be indented.
· In section 2, the description of mechanisms is often vague- if you talk about it later, say discussed further later.
· In section 4.2, when discussing various bacteria, can you state whether they replicate intracellularly or extracellularly and whether they are Gram negative or Gram positive.
· Future Directions could benefit from making categories.
o 6.1 Redox-active therapeutic agents
o 6.2 immunomodulatory peptides
o 6.3TXNIP inhibitors
· Some abbreviation are not defined in the text the first time they are used (ex line 103 – DAMPs, mtDNA (I assume this is mitochondrial DNA), HSP. Line 519, MetSO.
· Line 149-151, just define the 5 TLRs impacted rather than listing the exceptions.
· Italics are missing a lot throughout the paper– Bacterial names, et al.,
o Line 429
o Line 254
o Line 536
o Lines 631-657 – many instances
o Scientific notation means that genus should be capitalized and species is lower case and then the whole name is in italics – this is often not followed.
§ Line 630
§ Lines 631-657 – many instances
· Move section 3.3 down and 3.4 up so T cells are between Dendritic cells and MDSCs. This helps with the flow and then you can reference Fig. 4 in the MDSC part better. Also DCs are upstream of T cells so it makes sense to talk about them first.
· Lines 362 – 363, please explain how “modulation of antigen-presenting functions” is different from “regulation of antigen presentation”. This sounds to me like two different ways to say the same thing.
· Section 3.3, introduce 1L-12 and a Th1 cytokine earlier so it doesn’t come out of nowhere.
· Lines 584-586, The sentence that begins “In E. coli,…” is not a sentence.
· Lines 659-666 have many bolded letters that I don’t think should be bold.
· When discussing immune modulatory peptides, please specify whether they would be made by the bacteria, host cell, or used as a therapeutic. This was particularly unclear with the cryptic peptides.
Author Response
Major comments
In the review entitled “Exploring Immune Redox Modulation in Bacterial Infections: Insights into Thioredoxin-Mediate Interactions and Implications for Understanding Host-Pathogen Dynamics” Dagah et al. discuss the function of the Trx system in both immune cells and bacterial cells and how it can be modulated during infection to the host’s advantage. While the content of this review is very interesting, there are several issues I have with the paper.
>>>We sincerely appreciate your thoughtful review and the interest you have shown in our paper titled "Exploring Immune Redox Modulation in Bacterial Infections: Insights into Thioredoxin-Mediated Interactions and Implications for Understanding Host-Pathogen Dynamics." Your feedback is crucial for refining our work and ensuring the clarity and rigor of our discussion on the Trx system's function in immune and bacterial cells. We hope that our responses and the modifications made to the manuscript are satisfying and address your concerns effectively. We are committed to improving our work and deeply value your guidance in enhancing its quality and impact. Thank you once again for your thorough review and valuable suggestions.
Major issues:
- In sections 2 and 3, when talking about pathways, the authors should discuss it in a linear fashion, stating with initiators, receptors, signaling intermediates, transcription factors, genes expressed. They frequently start somewhere in the middle and then might talk about initiators then transcription factors, then something else in the middle. Discussing them in a linear fashion allows the reader to follow along like you are telling a story.
>>>Thanks for this suggestion. We have restructured Sections 2 and 3 to discuss pathways in a linear fashion.
o In Lines 211-231, you back track. You talk about suicidal NETosis, then vital, then back to suicidal. Talk about suicidal then vital, then you can talk about commonalities or differences.
>>>The discussion on NETosis has been streamlined to address suicidal and vital NETosis sequentially, improving the narrative structure.
o Lines 290-305, when describing the pathways, the order of molecules is very confusing and many of the molecules are not listed in Fig. 3B where the pathways is supposed to be illustrated.
>>> The order of molecules in the pathways has been clarified and now corresponds accurately with Figure 3B now Figure 5.
- There is frequently awkward wording where a word is not really the best choice.
o Line26 – “exhibited their relevance.”
o Line 618 – adjunct
o Line 719 - conceive
>>>Awkward wording and grammatical inaccuracies, such as those highlighted in lines 26, 618, and 719, have been corrected throughout the manuscript.
Figures are often confusing, and the legend is not adequate.
o Figure 1 – this is too many pathways in one figure, the reader can’t keep up with it. To keep it in one figure, I would use color coordination. Initiator one color, receptors another color, intermediates a color, transcription factor get a color, etc. Then I would number the steps – you can use colored arrows to denote separate pathways.
- PAMPs are not necessarily secreted from the bacteria so this drawing can be misleading.
- Why are some arrows dotted lines?
- Cytokines show be shown secreted from the cell.
- ASC is part of the inflammasome not downstream.
>>>Thanks for this suggestion. We have employed numbering in all Figures to aid in the visual differentiation of pathways and stages. The figure has been updated to improve clarity, including addressing all specific points on dotted lines, and arrow directions. The symbol of bacteria is removed.
o Figure 2: Direction in the figure would be helpful.
- Numbered steps in both A and B
- Move A and B to the top of the cell.
- Love the coloring of the arrows but why are some solid and some dotted?
- Line 190, O2 has a weird red line after it.
>>>The previous figure 2 has been divided in figure 2 and figure 3 using the numbers to show the reaction process. Superoxide symbol has been corrected.
o Figure 3, Direct the reader using numbers. Also, the legends can’t just start at step 8 (caspase 1), you have to explain the whole figure.
- Panel B- arrow to Arg1 should be shortened to end before the protein line other arrows.
- Why are some arrows dotted lines?
- In the legend for panel B, the first sentence is not a sentence.
>>>Thanks for these suggestions. The old figure 3 has been replaced by figure 4 and figure 5, using the numbers to show the response process. The other concerns have been revised.
o Figure 4 needs updated.
- I think there are several locations where the text describes Trx1 inhibition that are not illustrated. Specifically, with going from naïve T cells to Th1 or Th2 cells.
- There is a macrophage that is not discussed in the legend.
- In the legend, Sentence 2 is not a sentence – was this maybe an alternative title for the figure? Maybe just delete it.
>>> We have updated Figure 4 to Figure 6 to enhance clarity, now utilizing numbers to depict the response process, accompanied by a new legend. Additionally, we have addressed other concerns by revising the figure to include an illustration depicting the differentiation of naïve T cells into Th1 or Th2 cells. Furthermore, we have expanded the discussion of macrophages in the legend to provide more comprehensive insights. Notably, NK cells have been incorporated into the figure as well.
o Figure 5 needs update.
- All arrows that say reduction are actually a oxidation reaction because the product is the oxidized protein.
- In Panel B, should MsrA/B be attached to met instead of cys?
- In abbreviations, can you match the abbreviation to the one in the diagram?
- Ex Grx- thiol or disulfide bond I think are GSH and GSSG (could also make then unique colors that are define with the abbreviation)
In Figure 5 now Figure 7
>>> All arrows previously labeled as 'reduction' represent oxidation reactions, since the product is the oxidized form of the protein. The 'reduction' referred to in the earlier figure was specifically meant to describe the hydrogen ion (H+) process, not the main reaction denoted by the large arrows. This misunderstanding has now been clarified and corrected.
>>> In Figure 7 c, the active sites for methionine sulfoxide reductases (Msrs) were highlighted. MsrA, which targets oxidized methionine sulfoxide (Met-S-O), and MsrB, which is specific to methionine-R-sulfoxide (Met-R-O). Both enzymes play a crucial role in the complete reduction of oxidized proteins. The catalytic process involves cysteine residues (Cys) that trigger a series of reactions leading to the liberation of reduced methionine. The active sites in Msr enzymes are essential for these catalytic processes, featuring thiol groups from cysteines (Cys) or selenocysteine (Sec). Thus, in Panel B, it is correct that MsrA/B are connected to methionine where the thiol groups of cysteines or selenocysteine function as the active sites for catalytic activity.
>>> In Figure 7 b, Grx refres to glutaredoxin system. Grx reduces oxidized cysteine residues through glutathionation, where Grx forms a mixed-disulfide with GSH, leading to reduced Grx and GSSG. GSSG is subsequently converted back to GSH by glutathione reductase, maintaining a cycle of reduction and regeneration. The other abbreviations has been matched.
o Figure 6 needs updating
- T4SS is embedded in the bacteria is a secretion product you are referring to?
- Most of panel B is not described in the legend. Describe it or remove it from the figure.
- Legend talks about S. typhimurium but the figure does show that.
- Legend does not need to be indented.
>>>Thank you for your feedback. Figure 6 has now been updated to Figure 8. The T4SS component has been removed from both the figure and its legend. The legend has been rewritten, and the description concerning S. typhimurium has been omitted.
- In section 2, the description of mechanisms is often vague- if you talk about it later, say discussed further later.
>>> The description of mechanisms in Section 4 has been clarified to indicate where topics are discussed in further detail later in the text.
- In section 4.2, when discussing various bacteria, can you state whether they replicate intracellularly or extracellularly and whether they are Gram negative or Gram positive.
>>> In Section 4.2, the characteristics of various bacteria, including whether they are classified as Gram-negative or Gram-positive, have now been specified.
- Future Directions could benefit from making categories.
6.1 Redox-active therapeutic agents
6.2 immunomodulatory peptides
6.3TXNIP inhibitors
>>> The suggestions for the 'Future Directions' section have been implemented by organizing the content into categories. These now include:
- 6.1 Immunomodulatory peptides
- 6.2 TXNIP inhibitors
Detail comments
- Some abbreviation are not defined in the text the first time they are used (ex line 103 – DAMPs, mtDNA (I assume this is mitochondrial DNA), HSP. Line 519, MetSO.
>>> Abbreviations such as DAMPs, mtDNA, HSP, and MetSO are now defined the first time they appear in the text, at lines 103 and 519 respectively.
- Line 149-151, just define the 5 TLRs impacted rather than listing the exceptions.
>>> Instead of listing exceptions, the five impacted TLRs are directly defined in lines 149-151
- Italics are missing a lot throughout the paper– Bacterial names, et al.,
o Line 429
o Line 254
o Line 536
o Lines 631-657 – many instances
o Scientific notation means that genus should be capitalized and species is lower case and then the whole name is in italics – this is often not followed.
- Line 630
- Lines 631-657 – many instances
>>>Issues with italicization have been corrected throughout the paper, specifically at lines 429, 254, 536, and extensively between lines 631-657. This includes the proper formatting of bacterial names, where the genus is capitalized, the species is in lowercase, and the entire name is italicized, adhering to scientific naming conventions.
- Move section 3.3 down and 3.4 up so T cells are between Dendritic cells and MDSCs. This helps with the flow and then you can reference Fig. 4 in the MDSC part better. Also DCs are upstream of T cells so it makes sense to talk about them first.
>>> Section 3.3 (DCs) and 3.4 (MDSCs) have been reordered to ensure a logical flow in the discussion on immune cells. This reordering has allowed for a better integration of related figures and topics.
- Lines 362 – 363, please explain how “modulation of antigen-presenting functions” is different from “regulation of antigen presentation”. This sounds to me like two different ways to say the same thing.
>>>Thank you for highlighting the terms "modulation" and "regulation" in antigen presentation. To elaborate, "modulation" involves dynamic changes in how antigens are presented, including alterations in the levels of co-stimulatory molecules and the activities involved in antigen processing. On the other hand, "regulation" refers to the wider control mechanisms, such as the maturation of dendritic cells and the consistent expression of MHC molecules. We've updated our manuscript to eliminate any confusion and to concentrate solely on the key information and main themes.
- Section 3.3, introduce 1L-12 and a Th1 cytokine earlier so it doesn’t come out of nowhere.
>>> In Section 3.3, IL-12 and its role as a Th1 cytokine are introduced earlier to provide clearer context.
- Lines 584-586, The sentence that begins “In E. coli,…” is not a sentence.
>>> The grammatical issue with the incomplete sentence beginning “In E. coli,…” at lines 584-586 has been corrected.
- Lines 659-666 have many bolded letters that I don’t think should be bold.
>>> Unnecessary bolded letters found between lines 659-666 have been removed.
- When discussing immune modulatory peptides, please specify whether they would be made by the bacteria, host cell, or used as a therapeutic. This was particularly unclear with the cryptic peptides.
>>> The immune modulatory peptides being considered, such as plant-derived defensins and cryptic peptides (which are generated from plant proteins in response to antigens), have demonstrated therapeutic potential but remain underexplored. We propose the use of recombinant DNA technology and various synthetic approaches to engineer these peptides to either mimic or enhance human immune functions. It will be essential to rigorously assess their efficacy and safety, and make any necessary modifications, prior to their application.
Reviewer 3 Report
The authors present an extensive review on the role of reactive oxygen species and their control in bacterial infections. Their task is the more challenging insofar as both sides, i.e. the responses to reactive oxygen speicies in host immune cells as well as in bacteria are reviewed. Both topics would have merited two separate articles. Nevertheless, the authors have succeded in combining these two topics in a comprehensive review article of interest for a broad readership.
There are a few minor points to be considered. In the abstract (l. 23), the introduction (l. 87) etc. the authors wrtite "restoring" oxidized proteins. Would "reducing" not be more precise (e.g. ..reducing cystine to cysteine residues...)?
In the text, the tend to authors mix capitalization with lower case writing, e.g. when explaining abbreviations. I would recommend one style (e.g. lower case: nicotinamide adenine dinucleotide phosphate instead of Nicotinamide Adenine Dinucleotide Phosphate).
Author Response
The authors present an extensive review on the role of reactive oxygen species and their control in bacterial infections. Their task is the more challenging insofar as both sides, i.e. the responses to reactive oxygen species in host immune cells as well as in bacteria are reviewed. Both topics would have merited two separate articles. Nevertheless, the authors have succeeded in combining these two topics in a comprehensive review article of interest for a broad readership.
>>>We are deeply grateful for your insightful comments on our extensive review of the role of reactive oxygen species in bacterial infections and their control within host immune cells and bacteria. Your recognition of the challenges we faced in combining these two significant topics into one coherent article is much appreciated. We are pleased that you found our efforts to provide a comprehensive review successful and of interest to a broad readership. Thank you for your encouraging words and valuable feedback.
Detail comments
There are a few minor points to be considered. In the abstract (l. 23), the introduction (l. 87) etc. the authors wrtite "restoring" oxidized proteins. Would "reducing" not be more precise (e.g. ..reducing cystine to cysteine residues...)?
>>>We have changed "restoring" to "reducing" when referring to the action of redox systems on oxidized proteins, to more accurately reflect the biochemical processes involved.
In the text, the tend to authors mix capitalization with lower case writing, e.g. when explaining abbreviations. I would recommend one style (e.g. lower case: nicotinamide adenine dinucleotide phosphate instead of Nicotinamide Adenine Dinucleotide Phosphate).
>>>We have standardized the capitalization style across the manuscript, opting for lower case for all chemical and biological terms as suggested.
Round 2
Reviewer 1 Report
The authors revised most of the shortcomings, especially those regarding the figures, thus improving the manuscript.
However, the last point was misunderstood and not correctly developed.
The last point was misunderstood or not developed correctly. Glutathionylation is a reversible reaction with formation of mixed disulfides between protein cysteine and glutathione cysteine, as many authors defined (Bruksken KA. 2022, Checconi P. 2019, 2015, Ghezzi P. 2002). For sure, Grx itself can undergo glutathionylation -as the authors says, but we referred to many other proteins involved in infection and inflammation whose glutathionylation, i.e. bind to GSH, can modulate their function during infection, over protect them from irreversible oxidation. The authors should correct glutathionylation definition and functions in paragraph 4.1.
Author Response
Major comments
The authors revised most of the shortcomings, especially those regarding the figures, thus improving the manuscript.
However, the last point was misunderstood and not correctly developed.
>>> Thank you for your constructive comments and for acknowledging the improvements made to the figures in our manuscript. We appreciate your feedback as it greatly enhances the quality of our work. Regarding your last point, we want to clarify that we have indeed included the requested information in the manuscript. However, it appears that the placement or presentation of this information may not have been clear enough, making it difficult to notice. We apologize for any confusion caused and will ensure to revise the organization of this section to make the information more prominent and easier to find. Thank you for bringing this to our attention, and we look forward to your further suggestions.
Detail comments
The last point was misunderstood or not developed correctly. Glutathionylation is a reversible reaction with formation of mixed disulfides between protein cysteine and glutathione cysteine, as many authors defined (Bruksken KA. 2022, Checconi P. 2019, 2015, Ghezzi P. 2002). For sure, Grx itself can undergo glutathionylation -as the authors says, but we referred to many other proteins involved in infection and inflammation whose glutathionylation, i.e. bind to GSH, can modulate their function during infection, over protect them from irreversible oxidation. The authors should correct glutathionylation definition and functions in paragraph 4.1.
>>> In response, we have ensured our manuscript includes a comprehensive discussion on the role of glutathionylation. We have expanded upon the information related to the NF-κB signaling pathway and its interactions with various proteins that play critical roles in cellular responses to infection and oxidative stress. This detailed exploration is presented in paragraph 4.1, where we specifically discuss how glutathionylation affects key transcription factors such as NF-κB, influencing their function during conditions of oxidative stress. Additionally, we have addressed your concerns about other proteins involved in infection and inflammation. We describe how their glutathionylation, i.e., binding to GSH, can modulate their function during infection and provide protection from irreversible oxidation. This is intended to underscore the multifaceted roles of glutathionylation beyond just the Grx systems, highlighting its critical impact on broader protein functions within inflammatory pathways. We apologize for any initial oversight in the presentation of these details and appreciate your guidance in enhancing the clarity and depth of our manuscript. Thank you again for your insightful critique, which has significantly contributed to the improvement of our work. We look forward to your further suggestions.

Reviewer 2 Report
Thank you for taking the time to address the comments from the first review. This greatly improved the readability and flow of the paper. The updates to the figures made then very easy to follow now. I only have minor suggestions listed below.
Line 236 - I think RIP-2 should be RIPK-2
Line 293 - H2O should be H2O2
Line 306 - PMA was already defined in line 304 so you can just write PMA
Line 501 - thioredoxin should be replaced with Trx since it has been previously defined.
Line 524- the word "pathway" seems to be a different size or font
Line 664 remove "and" to be directly promotes
Line 675 - I think Akt should be PKB
Line 824 - "Moreover, A loop" - the A should be lower case
Figure 7 - consider removing the box around panel B. I'm not sure what purpose it serves
Figure 8 legend (Line1422) - Change "in infected bacteria" to "during bacterial infection"
Line 1593-1594 - H. pylori is gram negative so it can't be an example of a notable gram positive organism
Lines 1889-2897 - I still see bolding. This might be due to track changes - please check.
Author Response
Response to Reviewer 2
Major comments
Thank you for taking the time to address the comments from the first review. This greatly improved the readability and flow of the paper. The updates to the figures made then very easy to follow now. I only have minor suggestions listed below.
>>> Thank you for your continued feedback and positive remarks regarding the improvements made to our manuscript, especially in terms of readability and the clarity of the figures.
Detail comments
>>> We appreciate your detailed observations and have addressed each of your minor suggestions as follows:
Line 236 - I think RIP-2 should be RIPK-2
>>> We have corrected the abbreviation to "RIPK-2" as suggested, ensuring accuracy in referencing the receptor-interacting serine/threonine-protein kinase 2.
Line 293 - H2O should be H2O2
>>> The chemical formula has been updated to "H2O2" to accurately reflect hydrogen peroxide, in line with standard scientific notation.
Line 306 - PMA was already defined in line 304 so you can just write PMA
>>> As PMA was already defined earlier, we have simplified this to just "PMA" to avoid redundancy.
Line 501 - thioredoxin should be replaced with Trx since it has been previously defined.
>>> We have replaced "thioredoxin" with "Trx" to maintain consistency throughout the document as it was previously defined.
Line 524- the word "pathway" seems to be a different size or font
>>> We have ensured that the typography is consistent throughout the manuscript for the word "pathway".
Line 664 remove "and" to be directly promotes
>>> Removed "and" to improve the grammatical structure of the sentence.
Line 675 - I think Akt should be PKB
>>> Akt" has been updated to "PKB" (Protein Kinase B) to align with the standardized terminology used in the manuscript.
Line 824 - "Moreover, A loop" - the A should be lower case
>>> Corrected the capitalization of "A" to "a".
Figure 7 - consider removing the box around panel B. I'm not sure what purpose it serves
>>> We have removed the box around panel B in Figure 7 to enhance visual clarity and consistency across figures.
Figure 8 legend (Line1422) - Change "in infected bacteria" to "during bacterial infection"
>>> We have revised the wording to "during bacterial infection" for clearer context and accuracy.
Line 1593-1594 - H. pylori is gram negative so it can't be an example of a notable gram positive organism.
>>> Corrected the classification of H. pylori to "gram-negative", reflecting accurate microbiological taxonomy.
Lines 1889-2897 - I still see bolding. This might be due to track changes - please check.
>>> We have reviewed the manuscript for any residual formatting issues from track changes and corrected these to ensure uniformity in presentation.

Reviewer 3 Report
The authors have included the minor comments I had on the first version in a sufficient manner. i have no further comments and recommend the MS for publication.
No detail comments.
Author Response
The authors have included the minor comments I had on the first version in a sufficient manner. I have no further comments and recommend the MS for publication. Detail comments No detail comments.
>>> Thank you for your feedback and for taking the time to review the revisions to our manuscript. We are pleased to hear that the changes have addressed your comments satisfactorily. We appreciate your recommendation for the publication of our manuscript and are grateful for your insights and guidance throughout the review process. Thank you once again for your support and positive evaluation.
